# META-RESEARCHER: EMPOWERING PLANNING AND REFLECTION MECHANISMS IN LARGE REASONING MODELS FOR ADVANCED DEEP RESEARCH ABILITIES

## ABSTRACT

Deep research significantly reduces the time and cost of information gathering for researchers by collecting and integrating vast amounts of data. However, its uncontrollable planning and reflection phases during reasoning lead to errors or gaps in information collection, and make it challenging to ensure timely reflection for correcting and supplementing information—thereby performing suboptimally in complex tasks requiring extensive data gathering. To address this limitation, we propose Meta-Researcher, an **End-to-End Reinforcement Learning-based Deep Research Method** designed to equip Large reasoning models (LRMs) and non-reasoning models with metacognitive capabilities for autonomously executing the research process of "**Task Planning - Information Gathering - Process Reflection - Problem Solving**", thereby effectively tackling complex problems that require multiple rounds of information collection and reasoning. Firstly, our approach standardizes LRMs to explicitly output controllable planning and reflection processes rather than implicitly including them within reasoning, thus ensuring that LRMs demonstrate metacognitive abilities in practice. Secondly, we perform end-to-end optimization through the Group Relative Policy Optimization (GRPO) strategy to enhance the active decision-making capabilities of LRMs while strengthening the metacognitive process. Extensive experiments on two tasks — closed-ended question answering and open-ended topic research — demonstrate that Meta-Researcher significantly outperforms existing deep search methods, deep research methods, and proprietary systems. Our approach enhances the reliability and applicability of LRMs in complex task scenarios, offering a new paradigm for developing intelligent agents with autonomous research capabilities.

## 1 INTRODUCTION

Large Reasoning Models (LRMs) have shown impressive capabilities in various domains such as natural language understanding and scientific reasoning OpenAI (2024); Zhipeng et al. (2025). However, models relying solely on internal knowledge struggle with open-ended tasks that require complex information gathering and multi-step reasoning, often producing incomplete or inaccurate responses. This limitation can result in factual inaccuracies and hallucinated content, particularly in knowledge-intensive tasks Zhilin et al. (2018); Xanh et al. (2020), scenarios involving locally private information Shuting et al. (2024); Joohyun & Minji (2024), and time-sensitive applications Ofir et al. (2023b); Jie et al. (2024). Consequently, tightly coupling LRMs' reasoning mechanisms with external knowledge-acquisition tools (e.g., search engines and retrieval systems) has become a key research direction, as demonstrated by recent literature Shunyu et al. (2023); Jinhao et al. (2024).

Existing approaches for integrating LRMs with external information acquisition can be broadly classified into two categories: *(1) Retrieval-Augmented Generation (RAG)* Yunfan et al. (2023); Wenqi et al. (2024) and *(2) Treating search engines as external tools* Shunyu et al. (2023); Xiaoxi et al. (2025a). RAG-based methods prompt LRMs to generate query keywords for retrieving relevant passages from local or external databases, then incorporate the retrieved content into the model's context for further generation Akari et al. (2024a); Zhihong et al. (2023). An alternative paradigm integrates search engines as modular tools within LRMs' reasoning pipelines, enabling the model to perform multi-step searches and use end-to-end training to learn effective interaction strategies

autonomously Huatong et al. (2025b); Bowen et al. (2025); Huatong et al. (2025b). However, these methods primarily target closed-ended questions. Open-ended questions—lacking standard answers or allowing diverse, non-unique responses—present greater optimization challenges. This has spurred recent efforts in deep research OpenAI (2025a); Grok (2025); Gemini (2025).

The fundamental goal of deep research technology is to equip models with autonomous capabilities for exploring, extracting, and synthesizing vast external knowledge resources, ultimately producing high-quality response results. This methodology significantly reduces time expenditure and operational costs for researchers in knowledge-intensive fields during primary data acquisition. Existing open-source deep research methods typically rely on predefined workflows ByteDance (2025); JinaAI (2025), lacking the capability for dynamic optimization during the research process. Some studies Xiaoxi et al. (2025b); Yuxiang et al. (2025) have adopted reinforcement learning to mitigate these issues, effectively enhancing the model's capability for dynamic decision-making. However, they lack active control over task planning and autonomous reflection during reasoning. This often leads to collection errors or information gaps in data gathering, and struggle to dynamically identify these issues, which results in inefficiency or superficial conclusions when handling complex tasks.

To address the aforementioned challenges, we propose Meta-Researcher, an end-to-end reinforcement learning-based deep research method, which aims to enhance the LRM's metacognitive abilities, that is, the ability to actively identify information errors or gaps and iteratively supplement them. The key components of Meta-Researcher are structured as follows: *(1) Multi-step planning mechanism:* Decomposes complex research problems by decoupling them into a series of structured, simpler sub-questions, thereby reducing the difficulty of obtaining information for intricate issues. *(2) Multi-tool collaborative calling mechanism:* It progressively fills the information gaps in each sub-question through the coordinated invocation of multiple tools. *(3) Autonomous reflection mechanism:* Reflect in real time on information gaps, trigger iterative supplementary exploration, and further enrich external knowledge. *(4) Adaptive Question answering mechanism:* The framework employs differentiated strategies to generate tailored answers for distinct question types: closed-ended question answering and open-ended research. *(5) End-to-end reinforcement learning:* Employing GRPO to enhance the LRM's metacognitive capabilities, enabling it to autonomously execute the research process of "Task planning - Information gathering - Process reflection - Problem solving", thus upgrading from passive execution to active decision-making.

In summary, our core contributions are as follows:

- We proposed Meta-Researcher, a deep research method that makes uncontrollable planning and reflection processes controllable by explicitly outputting the planning and reflection mechanisms during reasoning—rather than implicitly keeping them in thinking content—thereby effectively addressing information collection errors and gaps in deep research.

- We developed an end-to-end reinforcement learning framework based on GRPO, which supports task planning, multi-round interleaved reasoning, collaborative invocation of multiple tools, and autonomous reflection for information gaps. Through carefully designed rewards, it further enhances LRMs' abilities in decision-making and metacognition, as well as their controllability.

- We validated the effectiveness of Meta-Researcher in closed-ended question answering and open-ended topic research tasks. It significantly outperforms existing methods, including deep search methods, iterative research approaches, and proprietary systems.

## 2 RELATED WORKS

**Retrieval-Augmented Generation.** Retrieval-Augmented Generation (RAG) enhances models' abilities by incorporating a retrieval mechanism, which provides rich external knowledge and mitigates the limitations of static parameters Patrick et al. (2020); Yujia et al. (2024a). Early RAG methods mainly included branch retrieval Jaehyung et al. (2024), summary generation Yucheng et al. (2023), and adaptive retrieval Soyeong et al. (2024). Recent studies have begun to focus on retrieval necessity Jiejun et al. (2024), query reformulation Xinbei et al. (2023); Liang et al. (2023), document compression Yujia et al. (2024b), and noise handling Guanting et al. (2025). In addition, some studies have manage to bridge both gaps between query inputs and generation targets, and those between document identifiers and answers through end-to-end training Akari et al. (2024b); Xiaoxi et al. (2024a;b). Recent studies have enhanced the problem-solving capabilities of RAG

systems by endowing them with autonomous decision-making abilities Prakhar et al. (2024); Zehui et al. (2025). However, existing RAG methods have yet to integrate the powerful reasoning capabilities of reasoning-oriented models, limiting their potential for further improving system performance on complex tasks.

**Search Engines Interaction.** Another approach to mitigating the limitations of models' static knowledge involves integrating search engines as external tools into the reasoning process. Early methods such as IRCoT Harsh et al. (2023) and ReAct Shunyu et al. (2023) employed prompting mechanisms to guide iterative reasoning and search engine invocation, while Toolformer Timo et al. (2023) leveraged supervised fine-tuning to enhance search capabilities. Given that R1-type models exhibit robust chain-of-thought reasoning capabilities, recent research Xiaoxi et al. (2025a) has leveraged prompt engineering to integrate retrieval and reasoning processes, encouraging models to autonomously explore more effective retrieval behaviors. To further integrate model reasoning with search engines, recent studies have begun exploring end-to-end RL to enhance the model's autonomous decision-making abilities and optimal timing for search engine employment. Bowen et al. (2025); Huatong et al. (2025b;a); Yuxiang et al. (2025); Shuang et al. (2025). Nevertheless, the application potential of this method in search engine invocation scenarios remains to be explored.

# 3 METACOGNITIVE CAPABILITIES OF LLMS

Metacognition is an important concept in psychology, referring to an individual's ability to recognize, monitor, and regulate their own cognitive processes. In simple terms, it is "thinking about thinking." Some studies suggest that the metacognitive capabilities of Large Language Models (LLMs) refer to their ability to identify, assess, and articulate the boundaries of their own knowledge Mark & Megan (2025). We propose a new interpretation of LLMs' metacognitive capabilities: the ability of models to autonomously recognize and reflect on the limitations of their existing knowledge, and **explicitly take a series of planning and reflective actions to continuously fill knowledge gaps**. Regardless of how these capabilities are defined, enhancing the metacognitive abilities of LLMs remains a focal point in their ongoing development.

# 4 METHODOLOGY: META-RESEARCHER

## 4.1 OVERVIEW OF THE META-RESEARCHER FRAMEWORK

As illustrated in Figure 1, the Meta-Researcher primarily consists of four key components and operates in two modes. The four key components are as follows:

**(1) Task Planning Component:** This component decomposes complex problems into multiple sets of executable simple subproblems step by step and explicitly displays the task plans in reasoning process. The motivation is that directly gathering information for complex original problems may result in fragmented or incomplete data. By decoupling the original problem, the model can focus on only one simple problem during each information collection process, thereby acquiring external information more comprehensively and accurately.

**(2) Tools Calling Component:** Meta-Researcher integrates Web Search Tools for information gathering. It aims to deeply fuse external tools with the model's reasoning process in a plug-and-play manner. Additional tools can be dynamically expanded based on actual needs.

**(3) Process Reflection Component:** To enable LRMs to autonomously identify knowledge boundaries, detect information gaps, and proactively trigger knowledge supplementation during reasoning, we introduce the Process Reflection Module. By explicitly outputting self-reflection process, rather than keeping it implicit within thinking process, we ensure it genuinely exhibits reflective behavior (although implicit reflection within thinking also confers reflective capacity, it remains uncontrollable and stochastic). Furthermore, reinforcement learning and a meticulously designed reward are employed to further enhance the metacognitive capabilities related to planning and reflection.

**(4) Question Answering Component:** Question answering component operates in two modes: Closed-Ended Problem-Solving Mode and Open-Ended Topic Exploration Mode. We design tailored reward functions for each response style and enable the model to automatically distinguish between question types—selecting the appropriate answering approach through end-to-end training.

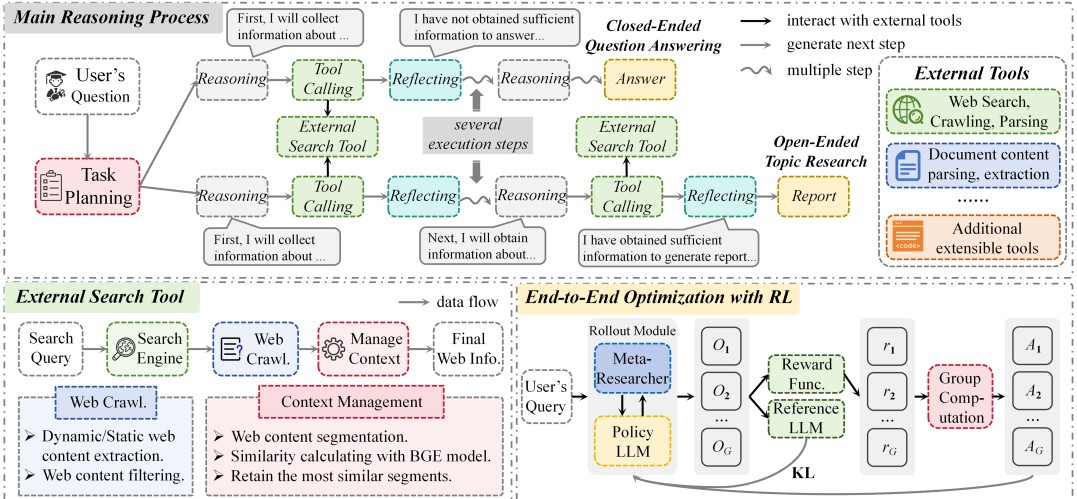

Figure 1: Overview of the Meta-Researcher framework. The framework consists of four key components: **Task Planning, Tools Calling, Process Reflection, and Question Answering**, operates in two modes: **Closed-Ended Question Answering and Open-Ended Topic Research**, with overall performance optimized through end-to-end reinforcement learning.

To ensure LRMs can explicitly output task planning and reflection processes, we designed the task planning module and the process reflection module as two virtual tools, leveraging the model's inherent tool-calling capability to guide it in invoking these tools at appropriate stages. Critically, the parameters for invoking these virtual tools are autonomously generated by the core model, and these parameters are precisely the outcomes of planning and reflection—ensuring that task planning and reflection capabilities are intrinsic properties of the model rather than externally imposed. Appendix E shows examples of using two virtual tools. The two operational modes are as follows:

**(I) Closed-Ended Question Answering Mode:** In this mode, Meta-Researcher is required to handle complex questions that exceed the internal knowledge of LRMs and generate a concise and accurate answer to precisely respond to users' inquiries. Given a specific task instruction $I$ and a query $q$, the LRM will generate a reasoning chain $\mathcal{R}$, which may be interspersed with calls to the Web Search Tool. The final output is usually a direct answer $\mathcal{A}$. The generation process is:

$$P(\mathcal{R}, \mathcal{A} \mid I, q) = \prod_{t=1}^{T_r} P(\mathcal{R}_t \mid \mathcal{R}_{<t}, I, q, \{\mathcal{O}_\tau^s\}_{\tau<t}) \cdot \prod_{t=1}^{T_y} P(\mathcal{A}_t \mid \mathcal{A}_{<t}, \mathcal{R}, I, q) \tag{1}$$

where $\mathcal{A}$ is answer sequence. $\{\mathcal{O}_\tau^s\}_{\tau<t}$ is outputs from Web Search calls before reasoning step $t$.

**(II) Open-Ended Topic Research Mode:** In this mode, the Meta-Researcher needs to comprehensively collect external information related to the research topic proposed by users from both breadth and depth dimensions. It then generates a detailed research report by integrating the external information with the reasoning capabilities of the LRM itself, thereby fully addressing the user's needs. The formal generation process follows a structure similar to Equation 1.

## 4.2 CONTEXT MANAGEMENT

Across diverse research topics, deep research processes generate extended reasoning trajectories and involve vast amounts of external information, causing the context window of LRMs to exceed capacity within just a few iterations. To mitigate this, we have developed a context management method specifically for web search tool and end-to-end reinforcement learning training process.

Specifically, for each webpage returned by search engine, we split its content based on common punctuation marks. Next, we use **BGE model** to compute the similarity between the search query and each segment. Finally, for each webpage, we retain only the segments most relevant to the search query. Additionally, when the length of inference context exceeds context window in training, we will perform a step-by-step truncation on tool return results within inference chain.

## 4.3 END-TO-END OPTIMIZATION WITH RL

To further enhance the model's metacognitive abilities—enabling LRMs to master autonomous decision-making and reflection on information gaps while deeply integrating external tool use with the model's reasoning process—we employ GRPO Shao et al. (2024) for end-to-end training. The algorithm samples a set of outputs $\{y_1, y_2, ..., y_G\}$ from the old poliy $\pi_{\theta_{old}}$ , and then optimizes the policy model by maximizing the following objective function:

$$\mathcal{J}(\theta) = \mathbb{E}_{q \sim D, \{y_i\}_{i=1}^G \sim \pi_{\theta_{old}}(\cdot|q)} \left[ \frac{1}{G} \sum_{i=1}^{G} \frac{1}{|y_i|} \sum_{t=1}^{|y_i|} \min(r_i^t A_i^t, \text{clip}(r_i^t, 1 - \epsilon, 1 + \epsilon)A_i^t) - \mathbb{KL} \right] \quad (2)$$

$$r_i^t = \frac{\pi_\theta(y_i^t|q, y_i^{<t})}{\pi_{\theta_{old}}(y_i^t|q, y_i^{<t})}, \qquad \mathbb{KL} = \beta D_{KL}(\pi_\theta || \pi_{\theta_{ref}}) \quad (3)$$

where $\epsilon$ and $\beta$ are hyperparameters, and $A_i^t$ is the advantage calculated from relative rewards within each group. We also adopt Mask-Based Loss Calculation in training Huatong et al. (2025b).

## 4.4 REWARD MODELING

To enable LRMs to stably and explicitly output planning and reflection processes—thereby effectively enhancing their metacognitive and autonomous decision-making abilities—we meticulously designed format rewards and answer rewards, adopting a two-stage training method.

### 4.4.1 FORMAT REWARDS.

**(1) Task Planning Phase:** Thinking process and the invocation of Task Planning Tool are encapsulated within <think> and <tool_call> tags, respectively, with the tool calling parameters representing the content of task planning. The format reward for this phase is as follows:

$$R_{plan} = \begin{cases} 1, & \text{if the task planning tool is called,} \\ 0, & \text{otherwise.} \end{cases} \quad (4)$$

**(2) Information Collection Phase:** Adopt the standard tool call format of the Qwen series models. The tool call process is encapsulated within <tool_call> tags and generates a JSON-formatted string. The format reward for the tool call stage is as follows:

$$R_{tool} = \begin{cases} 0.5, & \text{if the tool call format is correct,} \\ 0.0, & \text{otherwise.} \end{cases} \quad (5)$$

During training process, we count the total number of tool calls $n$ for each response. For every correctly formatted tool call, we increment $R_{tool}^{temp}$ by 0.5. The final format reward for the information-gathering phase is obtained by averaging the accumulated reward over $n$, i.e., $R_{tool} = R_{tool}^{temp}/n$.

**(3) Process Reflection Phase:** Similar to the task planning phase, thinking content and invocation of Process Reflection Tool are encapsulated within <think> and <tool_call> tags, respectively, with the tool calling parameters representing the content of self-reflection. The format reward is:

$$R_{ref} = \begin{cases} +0.1, & \text{if calling reflection tool and } 0 < s_r \le s_p, \\ -0.1, & \text{if calling reflection tool and } s_r > s_p, \\ -0.5, & \text{otherwise.} \end{cases} \quad (6)$$

where $s_r$ is the number of Reflection Tool calls in the current response, and $s_p$ is the number of steps in the task plan. If $0 < s_r \le s_p$, each Reflection Tool call will increase $R_{ref}$ by 0.1. If $s_r > s_p$, each additional Reflection Tool call will deduct 0.1 from $R_{ref}$. This mechanism encourages the use of as many reflection operations as possible within $s_p$ steps, while imposing an upper limit to prevent excessive reward accumulation. If Reflection Tool is not called at all, $R_{ref}$ will be set to –0.5.

**(4) Question Answering Phase:** Following the standard response style of the Qwen series models, the final reply is enclosed within the tags <answer>. The format reward is:

$$R_{ans} = \begin{cases} 0.5, & \text{if the answer format is correct,} \\ 0.0, & \text{otherwise.} \end{cases} \quad (7)$$

The final format reward is expressed as:

$$R_{format} = \begin{cases} R_{tool} + R_{ref} + R_{ans}, & \text{if } R_{plan} \text{ is } 1, \\ 0.0, & \text{if } R_{plan} \text{ is } 0. \end{cases} \tag{8}$$

### 4.4.2 ANSWER REWARDS.

We have designed two types of answer rewards respectively for the closed-ended question answering mode and the open-ended topic research mode.

**(1) Answer Reward for Closed-Ended Question Answering:** We extract the information within the `<answer>` tags from the response and employ a carefully designed scoring prompt to guide the large model in evaluating the predicted answer against the ground truth. A correct answer receives a score of 1, while an incorrect answer receives 0:

$$R_{answer} = \begin{cases} 1, & \text{if predicted answer is correct}, \\ 0, & \text{if predicted answer is incorrect}. \end{cases} \tag{9}$$

**(2) Answer Reward for Open-ended Topic Research:** The open-ended topic research mode focuses on exploratory questions that lack definitive answers, aiming to produce a thorough and insightful research report. Accordingly, we evaluate reports across four key dimensions: ***content completeness, information richness, consistency of data and facts, and structural rationality*** to determine the answer reward for this mode. For a detailed explanation of the four dimensions, please refer to the Appendix. Finally, we sum up the scores of the four dimensions, normalize them to the range of 0-1, and take this as $R_{answer}$.

### 4.4.3 THINKING LENGTH REWARD:

For reasoning models such as QwQ-32B, redundant thinking content can easily cause the generated output to exceed the context length, thereby disrupting the training process. To address this, we introduce a thinking length reward—a mechanism designed to encourage the model to reason fully within the length threshold while penalizing excessive thinking content. This approach helps the model learn to streamline its reasoning. The thinking length reward can be formalized as:

$$R_{think} = \begin{cases} + \delta \cdot len(think), & \text{if } len(think) \leq \eta, \\ - \delta \cdot len(think), & \text{if } len(think) > \eta. \end{cases} \tag{10}$$

where $\delta$ is a weight coefficient, $len(think)$ is the length of thinking in each round of response, and $\eta$ is the threshold.

### 4.4.4 TWO-STAGE TRAINING.

Due to the lack of intermediate annotations in the training data, the RL process is primarily driven by outcome-based rewards. To address this, we implement a two-phase reward mechanism.

Specifically, during the first phase, we train the model solely using format reward, without considering the correctness of its answers. This phase aims to ensure that the model truly possesses metacognitive abilities, transforming planning and reflective behaviors from uncontrollable to controllable, while simultaneously enhancing the stability of the model's interactions with external tools. The reward function for this phase is defined as follows:

$$Reward_{stage_1} = R_{format} \tag{11}$$

When the reward function in the first stage exceeds the preset threshold $\xi$, the training process advances to the second stage. In this phase, we adopt a hybrid training strategy that combines format rewards, answer rewards, and thinking length rewards. This approach aims to deeply integrate the model's reasoning process with the external tool invocation mechanism, and to refine task planning and process reflection, thereby further enhancing the model's autonomous decision-making capabilities and metacognitive processes. The second-stage reward function is defined as follows:

$$Reward_{stage_2} = R_{format} + R_{answer} + R_{think} - \xi \tag{12}$$

The meticulously designed reward mechanisms and training strategies ensure that Meta-Researcher can explicitly output task plans and process reflections, while improving its autonomous decision-making and metacognitive abilities.

Table 1: Comparison Results of Closed-Ended Question Answering Tasks. We report Pass@1 metric for all tasks. The best and second results are marked in **bold** and underlined, respectively. $^\dagger$ indicates results from their official releases. Results from larger or closed-sourced models are in gray color.

| Methods | GPQA (Science QA) | | | | GAIA (General AI Assist.) | | | | Bamboogle |
|---|---|---|---|---|---|---|---|---|---|
| | Phy. | Chem. | Bio. | Avg. | Level1 | Level2 | Level3 | Avg. | Avg. |
| *Direct Reasoning (w/o Retrieval)* | | | | | | | | | |
| Qwen2.5-32B | 52.3 | 30.1 | 68.4 | 43.4 | 20.5 | 9.6 | 8.3 | 13.6 | 60.8 |
| Deepseek-R1-32B | 82.5 | 41.9 | 73.7 | 62.6 | 23.1 | 17.3 | 8.3 | 18.4 | 59.2 |
| QwQ-32B | 81.4 | 39.8 | 68.4 | 60.6 | 30.8 | 15.4 | **25.0** | 22.3 | 56.0 |
| GPT-4o | 64.0 | 46.2 | 68.4 | 56.1 | 23.1 | 15.4 | 8.3 | 17.5 | 58.4 |
| DeepSeek-R1-671B | 90.7 | 57.0 | 84.2 | 74.2 | 43.6 | 26.9 | 8.3 | 31.1 | 68.8 |
| o1-preview$^\dagger$ | 89.4 | 59.9 | 65.9 | 73.3 | - | - | - | - | - |
| *Enhancing Reasoning with RAG Workflow* | | | | | | | | | |
| RAG-Qwen2.5-32B | 59.3 | 39.8 | 52.6 | 49.5 | 12.8 | 11.8 | 8.3 | 11.8 | 61.6 |
| RAgent-Qwen2.5-32B | 61.6 | 40.9 | 52.6 | 51.0 | 35.9 | 17.3 | 8.3 | 23.3 | 63.2 |
| RAG-QwQ-32B | 79.1 | 40.9 | 63.2 | 59.6 | 38.5 | 28.8 | 8.3 | 30.1 | 61.6 |
| RAgent-QwQ-32B | 81.4 | 43.0 | 68.4 | 62.1 | 51.2 | 26.9 | 8.3 | 33.9 | 64.0 |
| *Autonomous Search within Reasoning* | | | | | | | | | |
| Search-o1-QwQ | 84.9 | 49.5 | 73.6 | 67.2 | 53.8 | 34.6 | 16.7 | 39.8 | 67.2 |
| Search-R1-QwQ | 89.5 | 50.5 | 73.6 | 69.6 | 51.3 | 40.4 | 16.7 | 41.7 | 69.6 |
| R1-Searcher-QwQ | 88.3 | 51.6 | 73.6 | 69.6 | 53.8 | 44.2 | 16.7 | 44.6 | 74.4 |
| WebThinker-QwQ-RL | 90.7 | 50.5 | **78.9** | 70.7 | 56.4 | **50.0** | 16.7 | 48.5 | 74.4 |
| Meta-Researcher-QwQ-Base | 83.7 | 45.2 | 68.4 | 68.1 | 48.7 | 40.4 | 8.3 | 39.8 | 71.2 |
| Meta-Researcher-QwQ-RL | **93.0** | **53.7** | **78.9** | **73.2** | **61.5** | 48.1 | **25.0** | **50.5** | **79.2** |

## 5 EXPERIMENTS

### 5.1 SCENARIOS AND DATASETS

We evaluated Meta-Researcher's performance in two scenarios:

**Closed-Ended Question Answering:** This scenario evaluates the model's ability to solve closed-ended problems. We adopted the following benchmark datasets: **GPQA** David et al. (2023); **GAIA** Grégoire et al. (2024); **Bamboogle** Ofir et al. (2023a); and **Humanity's Last Exam (HLE)** Long et al. (2025). In this scenario, we employ Qwen2.5-72B-Instruct model and evaluate method performance using the general closed-ended evaluation method.

**Open-ended Topic Research:** This scenario primarily evaluates the model's ability to conduct research on open-ended questions. We adopt the **glaiveai/reasoning-v1-20m (Glaive)** Glaive (2025) dataset and follow the methodology proposed by Xiaoxi et al. (2025b), using only 30 selected samples as the test set. For the sake of fairness, we adopt the evaluation method in Xiaoxi et al. (2025b) instead of the open-ended report scoring system we designed for training to assess performance.

### 5.2 BASELINES

**Direct Reasoning:** Models using only static parametric knowledge without retrieval. Includes open-sourced models (Qwen2.5-32B-Instruct, Deepseek-R1-Distill-Qwen-32B, QwQ-32B, Deepseek-R1-671B) and closed-source models (GPT-4o, o1-preview OpenAI (2024), Gemini-2.0-Flash-Thinking Gemini (2025), o3-mini OpenAI (2025b)).

**Retrieval-Augmented Reasoning:** Methods using external knowledge from search engines. We test two approaches: **(1) Standard RAG**, which calls the search engine only once before generating answers. **(2) RAgent**, a retrieval-augmented agent capable of iteratively calling the search engine multiple times to retrieve external information.

**Autonomous Search within Reasoning:** Methods integrating the search engine as an external tool into the reasoning process. We compared open-source systems such as Search-o1 Xiaoxi et al.

Table 2: Main results on open-ended topic research tasks.

| Methods | Glaive (General Research Tasks) | | | | |
|---|---|---|---|---|---|
| | Comp. | Thorough. | Fact. | Coherence | Avg. |
| *Retrieval-Augmented Report Generation* | | | | | |
| RAgent-QwQ-32B | 5.5 | 5.4 | 6.3 | 6.1 | 5.8 |
| RAgent-DeepSeek-R1 | 6.6 | 6.4 | 7.1 | 7.1 | 6.8 |
| *Deep Research Systems* | | | | | |
| Grok3 DeeperSearch | 6.4 | 6.1 | 7.0 | 6.5 | 6.5 |
| Deer-Flow | 7.8 | 7.8 | **7.8** | 7.9 | 7.8 |
| WebThinker-QwQ-RL | **8.2** | **8.3** | 7.7 | 7.9 | **8.1** |
| *Autonomous Plan-Execute-Reflect-and-Write (Ours)* | | | | | |
| Meta-Researcher-QwQ-Base | 6.7 | 6.6 | 7.3 | 7.2 | 7.0 |
| Meta-Researcher-QwQ-RL | 8.0 | 7.9 | **7.8** | **8.1** | 7.9 |

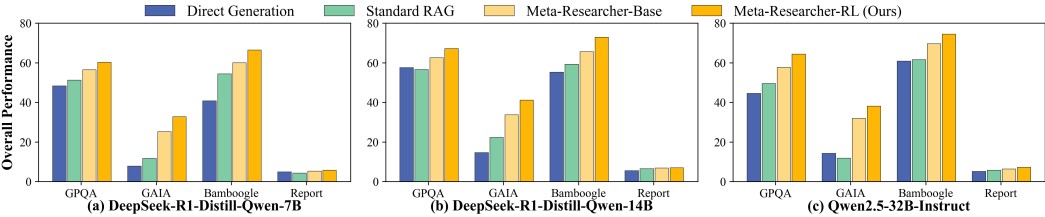

Figure 2: Performance with reasoning models across different sizes and non-reasoning models.

(2025a), Search-R1 Bowen et al. (2025), R1-searcher Huatong et al. (2025b), WebThinker Xiaoxi et al. (2025b), and Deer-Flow ByteDance (2025), as well as proprietary systems like OpenAI Deep Research OpenAI (2025a), and Grok3 DeeperSearch Grok (2025).

## 5.3 RESULTS ON CLOSED-ENDED QUESTION ANSWERING

Table 1 presents the performance of Meta-Researcher in the closed-ended question answering scenario. Based on the results, we draw the following conclusions:

**(1) Limitations of direct reasoning and RAG workflow:** Relying solely on the model's direct reasoning ability leads to poor performance in knowledge-intensive tasks (e.g., GAIA and Bamboogle), highlighting the limits of static knowledge. Deepseek-R1-671B and o1-preview perform better on GPQA due to relevant pre-training knowledge. Although the RAG workflow provides some improvement for such problems, it still performs poorly on complex issues due to insufficient integration between reasoning capabilities and external information retrieval.

**(2) Advantages of autonomous searching in reasoning processes:** Methods such as Search-o1, Search-R1, and R1-Searcher, which integrate autonomous searching into reasoning, demonstrate significant advantages over direct reasoning and RAG workflows. This improvement is particularly evident in complex, knowledge-intensive tasks like GAIA and Bamboogle, showing that combining the model's reasoning capabilities with external information retrieval can effectively mitigate subpar responses caused by insufficient static parametric knowledge.

**(3) Advancement of the Meta-Researcher framework:** Our Meta-Researcher framework has further improved performance, achieving better results across all benchmark datasets. Notably, compared to the untrained base version (Meta-Researcher-base), the Meta-Researcher framework optimized through reinforcement learning demonstrates a significant performance improvement. This proves that reinforcement learning can effectively enhance the model's autonomous decision-making and metacognitive abilities. Additional comparative results are provided in Appendix.

## 5.4 RESULTS ON OPEN-ENDED TOPIC RESEARCH

Table 2 presents the performance of Meta-Researcher on the Glaive scientific report generation task. It evaluates the performance of different methods across four dimensions: Completeness(Comp.), Thoroughness(Thorough.), Factuality(Fact.), and Coherence. Meta-Researcher achieved the second-

Table 3: Ablation studies of Meta-Researcher.

| Methods | Benchmarks for two scenarios | | | | |
|---|---|---|---|---|---|
| | GPQA | GAIA | Bam. | HLE | Glaive |
| **Meta-Researcher-QwQ-RL** | **73.2** | **50.5** | **79.2** | **18.4** | **7.9** |
| w/o Plan | 72.2 | 45.6 | 75.2 | 16.8 | 6.7 |
| w/o Reflect | 71.7 | 45.6 | 73.6 | 16.2 | 7.2 |
| w/o Plan & Reflect | 69.6 | 43.6 | 70.4 | 14.6 | 5.9 |

highest overall score (**7.9**), outperforming not only RAG baselines but also leading advanced deep research systems such as Deer-Flow. These results demonstrate the effectiveness of the "**task planning - information gathering - process reflection - problem solving**" strategy, which effectively leverages the dynamic planning and information gap-filling capabilities of LRMs. In addition, Web-Thinker ranks first due to its powerful report drafting tools specifically designed for report writing.

## 5.5 APPLICABILITY OF META-RESEARCHER

To evaluate the adaptability of Meta-Researcher across reasoning models of different scales and non-reasoning models, we conducted experiments on DeepSeek-R1-Distill-Qwen models of varying sizes (7B and 14B) as well as Qwen2.5-32B-Instruct model, as illustrated in Figure 2. To ensure inference stability for both R1-based LRMs and non-reasoning models, we employed a cold-start supervised fine-tuning method. Specifically, we fine-tuned the models using 6,000 reasoning trajectories collected by QwQ-32B-based Meta-Researcher, followed by reinforcement learning training.

In experiments involving both models of different scales and non-reasoning models, those trained using the Meta-Researcher method consistently outperformed direct inference, standard RAG baselines, and untrained methods across all four tasks (GPQA, GAIA, Bamboogle, and Glaive). These results demonstrate that the Meta-Researcher framework is highly effective and widely applicable in enhancing the deep research capabilities of various language models.

## 5.6 ABLATION STUDIES

We evaluated the contribution of core components of Meta-Researcher through ablation studies, with the results summarized in Table 3. **(1) Importance of Task Planning Component:** Task planning for complex problems, along with the iterative execution of information collection, significantly improved question answering performance, confirming the effectiveness of this approach. **(2) Effectiveness of Process Reflection Component:** When evaluating Meta-Researcher (based on QwQ-32B) on benchmarks like GPQA—which do not require extensive information gathering—removing the process reflection component results in only a small drop in performance. However, in benchmarks demanding iterative, complex information retrieval (e.g., GAIA and Bamboogle), strengthening the reflection process significantly improves solution effectiveness. This demonstrates that self-reflection is effective in fully gathering information and filling information gaps. **(3) Advancement of Meta-Researcher:** When both planning and reflection component are disabled, the method exhibits significant degradation in handling closed-ended questions and open-ended research tasks. This indicates that endowing LRMs with controllable metacognitive abilities and enhancing them through training holds significant value for complex knowledge-intensive tasks.

## 6 CONCLUSIONS

Meta-Researcher empowers LRMs to autonomously execute the research process of "Task Planning - Information Gathering - Process Rflection - Problem Solving". By leveraging reinforcement learning, it significantly enhances the models' **autonomous decision-making and metacognitive abilities**, effectively addressing the limitations of both reasoning and non-reasoning models when handling complex problems requiring multi-round information collection and inference. Large-scale experiments demonstrate that Meta-Researcher not only consistently outperforms existing methods but also surpasses powerful proprietary systems. These findings highlight its potential to substantially advance language models' deep research capabilities. Moreover, they provide critical insights for developing more powerful and flexible deep research systems.

REPRODUCIBILITY STATEMENT

To ensure the reproducibility of the results presented in this paper, we have organized and made publicly available all key supporting materials in the **Appendix**: Appendix B provides detailed descriptions of the experimental datasets and implementation details. Appendix D presents the complete prompt designs used in the proposed method. Furthermore, the source code is provided in **supplementary materials**, enabling full reproducibility of the experimental results.

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

## A  THE USE OF LARGE LANGUAGE MODELS (LLMS)

During the writing process of this paper, we utilized the large language models (LLMs) as an auxiliary tool, solely for polishing the grammar and expression of the paper to enhance its standardization and readability. The research conception, core content, experimental design, and conclusions of the paper were all independently completed by the authors, with the large language models not participating in any substantive aspects of the research or creative process.

## B  EXPERIMENTAL DETAILS

### B.1  DATASETS

#### B.1.1  TESTING DATA

We evaluate Meta-Researcher's capacity for closed-ended question answering and open-ended topic research on five widely adopted datasets:

- **GPQA David et al. (2023):** A graduate-level scientific Q&A corpus covering sub-disciplines of biology, physics, and chemistry—including quantum mechanics, organic chemistry, and molecular biology. The 198 questions in the diamond partition were authored by 61 domain experts holding or pursuing PhDs, and each item underwent a rigorous validation pipeline to ensure exceptional difficulty and quality. We used the diamond set consisting of 198 questions.

- **GAIA Grégoire et al. (2024):** A benchmark dataset specifically designed to evaluate General Artificial Intelligence (AGI) capabilities, consisting of real-world problems that require a range of fundamental abilities such as reasoning, multimodal processing, web browsing, and proficiency in using general tools. Since our method only targets text-based questions and cannot handle other modalities, we use the text-only validation subset, comprising 103 questions.

- **Bamboogle Ofir et al. (2023a):** A compact, human-curated dataset spanning diverse domains and featuring uniquely phrased question types. It deliberately compiles complex queries that Google fails to answer correctly, providing a rigorous benchmark for evaluating models' compositional reasoning across heterogeneous fields.

- **Humanity's Last Exam (HLE) Long et al. (2025):** A multimodal benchmark at the forefront of human knowledge, designed to be the ultimate closed academic benchmark with extensive disciplinary coverage within its category. This dataset consists of 2,500 challenging questions spanning fields such as mathematics, humanities and social sciences, and natural sciences, all accompanied by clear and easily verifiable answers. Due to the large scale of HLE, we use a text-only subset of 500 randomly selected items from the work of Xiaoxi et al. (2025b) as the experimental data.

- **glaiveai/reasoning-v1-20m (Glaive) Glaive (2025):** A large-scale reasoning dataset released by Glaiveai in 2025, containing approximately 20 million reasoning trajectories covering complex problems in various fields such as mathematics, programming, and science. This dataset aims to help models learn complex reasoning logic and improve their performance in multi-step reasoning tasks by providing abundant examples of reasoning processes. Following the setup of Xiaoxi et al. (2025b), we used only 30 instances from it as the test set.

### B.1.2 TRAINING DATA

During the reinforcement learning training phase, we use the following datasets to enhance the performance of Meta-Researcher in both closed-ended question answering and open-ended subject research modes:

- **ORION Lisheng et al. (2025):** A benchmark for open-web reasoning evaluation specifically designed based on long-tail entities. Unlike existing datasets that focus on high-frequency topics, ORION emphasizes reasoning over less common entities across ten diverse domains. It covers both English and Chinese samples and contains 310 annotated examples, each linked to authoritative sources. We use this dataset to enhance Meta-Researcher's capability in closed-ended question answering.

- **OpenResearchBench:** An open-topic research training dataset manually constructed by us, specifically designed to enhance the report generation capabilities of LRMs. This dataset includes a total of 300 Chinese and English open-topic research questions across 15 scenarios, such as consumer decision-making research, interpretation of news hotspots, travel guides, and consumption guides, with no standard answers provided.

When verifying the applicability of Meta-Researcher, to ensure the stability of the reasoning process in DeepSeek-R1-Distill-Qwen models of varying sizes (7B and 14B) and the Qwen2.5-32B-Instruct model, we conducted cold-start supervised fine-tuning using 6k reasoning trajectories collected by the QwQ-32B-based Meta-Researcher. The original questions for these 6k reasoning trajectories are all derived from the SuperGPQA benchmark:

- **SuperGPQA Team et al. (2025):** Designed to evaluate the knowledge and reasoning abilities of large language models (LLMs) across 285 graduate-level disciplines, SuperGPQA contains 26,529 questions covering 13 subjects, 72 fields, and 285 disciplines, with at least 50 questions per discipline. Encompassing a broad range of graduate-level topics, it is intended to serve as a challenging frontier for LLM evaluation.

### B.2 IMPLEMENTATION DETAILS

In the construction and training of Meta-Researcher, we adopted the open-source QwQ-32B as base model, with a temperature of 0.7, a top_p value of 0.8, and a top_k value of 20. In the experiment, we only used the search tool, employing the Google Web Search API configured for the US-EN region, retrieving the top 10 results for each query. The web content corresponding to the URLs was then crawled using Crawl4AI UncleCode (2024). In Context Management Module, the bge-reranker-v2-m3 model is used to calculate similarity, with each chunk size set to 512. The maximum number of task planning steps {max_step_num} is set to 5. Moreover, during the reward calculation process in the Process Reflection Phase, we define $s_p$ as the actual number of steps planned in the current response planning process. The thinking length reward weight coefficient $\delta$ is set to 1e-4, and the thinking length threshold $\eta$ is set to 2048. In Two-Stage Training Process, the threshold $\xi$ for first stage is set to 1.0. Model training consists of 10 iterations of GRPO with a maximum sequence length of 40960, a learning rate set to 2e-7, a batch_size of 64 and a group size of 5. All experiments are conducted on 4 nodes of 8 NVIDIA H100-80GB GPUs.

Table 4: Comparison Results of Closed-Ended Question Answering Tasks on Humanity's Last Exam Benchmark. We report Pass@1 metric for all tasks. The best and second results are marked in **bold** and underlined, respectively. [†] indicates results from their official releases. Results from larger or closed-sourced models are in gray color for reference.

| Methods | Humanity's Last Exam (Extremely Hard Reasoning Tasks) | | | | | | | | |
| --- | --- | --- | --- | --- | --- | --- | --- | --- | --- |
| | Math | Bio/M | Physic | CS/AI | Human | Chem. | Engine | Other | Avg. |
| *Direct Reasoning (w/o Retrieval)* | | | | | | | | | |
| Qwen2.5-32B | 6.0 | 7.0 | 2.0 | 3.2 | 8.0 | 6.7 | 5.3 | 4.4 | 5.4 |
| Deepseek-R1-32B | 7.4 | 9.3 | 4.0 | 4.8 | 8.0 | 6.7 | 5.3 | 6.7 | 6.8 |
| QwQ-32B | 12.6 | 14.0 | 4.0 | 7.9 | 6.0 | 13.3 | 5.3 | 4.4 | 9.6 |
| GPT-4o[†] | 2.4 | 5.3 | 2.2 | 1.2 | 2.9 | 1.9 | 1.3 | 3.5 | 2.6 |
| DeepSeek-R1-671B[†] | 9.3 | 8.6 | 5.8 | 7.4 | 11.0 | 5.6 | 10.3 | 7.5 | 8.6 |
| Gemini-2.0-Flash-Thinking[†] | 8.5 | 7.4 | 5.3 | 5.8 | 7.1 | 6.5 | 3.8 | 4.0 | 7.1 |
| o3-mini (High)[†] | 18.8 | 11.1 | 14.2 | 11.1 | 6.2 | 10.2 | 7.7 | 8.0 | 14.0 |
| *Enhancing Reasoning with RAG Workflow* | | | | | | | | | |
| RAgent-Qwen2.5-32B | 6.5 | 7.0 | 4.0 | 3.2 | 8.0 | 13.3 | 5.3 | 4.4 | 6.0 |
| RAgent-QwQ-32B | 11.2 | 16.3 | 6.0 | 4.8 | 10.0 | 13.3 | 10.5 | 8.9 | 10.0 |
| *Autonomous Search within Reasoning* | | | | | | | | | |
| OpenAI Deep Research[†] | - | - | - | - | - | - | - | - | 26.6 |
| Search-o1-QwQ | 12.1 | 11.6 | 2.0 | 7.9 | 14.0 | 6.7 | 10.5 | 15.6 | 10.8 |
| Search-R1-QwQ | 16.3 | 23.3 | 4.0 | 9.5 | 16.0 | 20.0 | **15.8** | 11.1 | 14.4 |
| R1-Searcher-QwQ | 16.7 | 18.6 | 2.0 | 11.1 | 14.0 | 13.3 | 10.5 | 8.9 | 13.4 |
| WebThinker-QwQ-RL | 16.7 | **25.6** | 2.0 | 12.7 | 18.0 | **26.7** | **15.8** | 15.6 | 15.8 |
| Meta-Researcher-QwQ-Base | 15.3 | 18.6 | 2.0 | 9.5 | 14.0 | 20.0 | **15.8** | 11.1 | 13.2 |
| Meta-Researcher-QwQ-RL | **19.5** | **25.6** | **8.0** | **15.9** | **20.0** | **26.7** | **15.8** | **17.8** | **18.4** |

# C  ADDITIONAL EXPERIMENTAL RESULTS

## C.1  EFFECTIVENESS OF META-RESEARCHER

To comprehensively validate the effectiveness of Meta-Researcher in closed-domain question-answering scenarios, we conducted supplementary comparative experiments on the HLE benchmark, with results detailed in Table 4. On the HLE dataset, Meta-Researcher again outperformed all baseline methods, achieving a score of **18.4**, further confirming the effectiveness of our approach. Notably, even the untrained base version of Meta-Researcher performs better than methods such as Search-o1, indicating that task planning and reflection processes hold significant value in reasoning and solving complex problems. Importantly, compared to the untrained base version, the model exhibits a significant performance improvement after end-to-end reinforcement learning training (a relative improvement of 39.4%), validating that our designed end-to-end training strategy can effectively enhance the metacognitive capabilities of Logical Reasoning Models (LRMs).

## C.2  ABLATION STUDIES OF TWO-STAGE TRAINING

To elucidate the specific benefits of each training stage, we conducted additional experiments, the results are presented in Table 5.

From the experimental results, we observe that when training solely with format-based rewards (+ only stage1 Training), the model's performance plateaus at the 14-th step, achieving only a marginal improvement over the untrained baseline model. This indicates that merely enforcing a specific format does not significantly enhance the model's reasoning capabilities. In contrast, the model trained exclusively with second-stage rewards (+ only stage2 Training) demonstrates performance comparable to that of the complete two-stage approach, albeit with a slower convergence rate.

This indicates that the primary performance enhancement of the model stems from its strengthened planning, reflection, and reasoning capabilities, which are reinforced through the answer-based reward mechanism. Additionally, initially training the model to adhere to a structured format facili-

Table 5: Ablation Studies of Two-stage Training. $ACC_L$ denotes the performance when adopting LLM-as-Judge. $Step$ indicates the number of training steps required to reach this performance.

| Methods | GPQA | | GAIA | | Bamboogle | | HLE | |
|---|---|---|---|---|---|---|---|---|
| | $ACC_L$ | $Step$ | $ACC_L$ | $Step$ | $ACC_L$ | $Step$ | $ACC_L$ | $Step$ |
| Meta-Researcher-32B-base | 68.1 | 0 | 39.8 | 0 | 71.2 | 0 | 13.2 | 0 |
| + only stage1 Training | 69.1 | 14 | 42.7 | 14 | 73.6 | 14 | 14.0 | 14 |
| + only stage2 Training | **73.2** | 70 | 48.5 | 70 | **79.2** | 70 | 18.2 | 70 |
| + Two-stage Training | **73.2** | 54 | **50.5** | 54 | **79.2** | 54 | **18.4** | 54 |

tates faster convergence. The first stage focuses on ensuring format compliance, while the second stage refines the model's planning, reflection, and reasoning abilities. These two stages serve distinct yet complementary roles, working together to help the model identify optimization directions more effectively than a mixed training approach that combines format and answer rewards.

### C.3 ABLATION STUDIES OF DIFFERENT REWARD COMPONENTS

To evaluate the effectiveness and importance of different reward components, we conducted ablation experiments, with the results shown in Table 6.

The results show that for closed-ended evaluation datasets: (1) Answer Reward contributes most significantly to Meta-Researcher's performance. Its removal leads to a notable decline in performance. (2) Format Reward ranks second in importance, as it encourages more frequent model reflections, enabling more comprehensive information supplementation and collection. (3) Thinking Length Reward has the least impact when removed, particularly on the GPQA dataset. Since GPQA inherently requires fewer search iterations, context lengths remain short, making truncation due to excessive length unlikely even without this reward. However, on the GAIA and Bamboogle datasets, restricting the thinking length can prevent the generation of overly long responses, thereby improving performance.

Table 6: Ablation Studies of different reward components. $ACC_L$ denotes the performance when adopting LLM-as-Judge. *w/o* indicates the removal of a specific reward.

| Methods | GPQA | GAIA | Bamboogle | HLE | Glaive |
|---|---|---|---|---|---|
| | $ACC_L$ | $ACC_L$ | $ACC_L$ | $ACC_L$ | $ACC_L$ |
| Meta-Researcher-32B-RL | **73.2** | **50.5** | **79.2** | **18.4** | **7.9** |
| *w/o* format reward | 71.7 | 45.6 | 77.6 | 17.0 | 6.5 |
| *w/o* thinking length reward | 73.2 | 47.5 | 77.6 | 18.2 | 6.2 |
| *w/o* answer reward | 69.1 | 42.7 | 73.6 | 14.0 | 7.2 |

For open-ended evaluation datasets, thinking length reward and format reward are particularly crucial. Open-ended scenarios require comprehensive information collection and complete report generation. Removing the thinking length reward can lead to excessively long contexts, resulting in incomplete reports and severely degraded performance. Similarly, poor formatting can disrupt planning, reflection, and search processes, ultimately compromising the comprehensiveness of information collection.

### C.4 IMPACT OF REINFORCEMENT LEARNING ON AGENT BEHAVIOR

To analyze the impact of reinforcement learning (RL) on the behavior of Meta-Researcher, we analyzed the actual number of search tool invocations and the performance of Meta-Researcher before and after reinforcement learning training on three test sets: GPQA, GAIA, and Bamboogle. The results are summarized in Table 7, where Meta-Researcher-32B-base represents the model without reinforcement learning training. It is worth mentioning that RL training did not lead to a significant increase in the number of search rounds; instead, the observed performance improvements primarily stemmed from enhancements in the model's planning, reflection, and reasoning capabilities during the training process.

Table 7: Impact of Reinforcement Learning on Agent Behavior. $ACC_L$ denotes the performance when adopting LLM-as-Judge. $S_{Turn}$ refers to the actual count of search tool invocations.

| Methods | GPQA | | GAIA | | Bamboogle | |
|---|---|---|---|---|---|---|
| | $ACC_L$ | $S_{Turn}$ | $ACC_L$ | $S_{Turn}$ | $ACC_L$ | $S_{Turn}$ |
| Meta-Researcher-32B-Base | 68.1 | 0.9 | 39.8 | 3.6 | 71.2 | 3.4 |
| Meta-Researcher-32B-RL | **73.2** | 1.6 | **50.5** | 4.7 | **79.2** | 4.2 |

Meanwhile, the Tongyi team, in their research on WebSailor Li et al. (2025), pointed out that first constructing cold-start data for fine-tuning the model and then conducting reinforcement learning can effectively enhance the model's multi-round search capability, thereby achieving superior results. In their experiments, the cold-start model consistently maintained a high and stable level of tool invocation throughout the entire reinforcement learning training period. In contrast, the model subjected to direct reinforcement learning, despite experiencing a continuous increase in invocation frequency, still demonstrated a significantly lower overall level. This indicates that relying solely on autonomous exploration through reinforcement learning makes it difficult for the model to master complex operational patterns such as ultra-multi-round searches, a conclusion that aligns with our experimental findings.

## C.5 WALL-CLOCK TIME COMPARISON ACROSS METHODS

We compare the wall-clock time consumption of direct reasoning, enhancing reasoning with RAG workflow, and autonomous search within reasoning. The results are summarized in Table 8. Given that direct reasoning and enhanced reasoning do not require reinforcement learning training, we solely compared the wall-clock time of various methods during the actual execution phase.

Table 8: Comparison of the Wall-Clock Time Among Different Methods.

| Methods | GPQA (Science QA) | | | | GAIA (General AI Assist.) | | | | Bam. | Time |
|---|---|---|---|---|---|---|---|---|---|---|
| | Phy. | Chem | Bio. | Avg. | Level | Level2 | Level3 | Avg. | Avg. | - |
| *Direct Reasoning (w/o Retrieval)* | | | | | | | | | | |
| Qwen2.5-32B | 52.3 | 30.1 | 68.4 | 43.4 | 20.5 | 9.6 | 8.3 | 13.6 | 60.8 | 5.7s |
| Deepseek-R1-32B | 82.5 | 41.9 | 73.7 | 62.6 | 23.1 | 17.3 | 8.3 | 18.4 | 59.2 | 16.1s |
| QwQ-32B | 81.4 | 39.8 | 68.4 | 60.6 | 30.8 | 15.4 | **25.0** | 22.3 | 56.0 | 14.7s |
| *Enhancing Reasoning with RAG Workflow* | | | | | | | | | | |
| RAG-Qwen2.5-32B | 59.3 | 39.8 | 52.6 | 49.5 | 12.8 | 11.8 | 8.3 | 11.8 | 61.6 | 9.8s |
| RAgent-Qwen2.5-32B | 61.6 | 40.9 | 52.6 | 51.0 | 35.9 | 17.3 | 8.3 | 23.3 | 63.2 | 11.6s |
| RAG-QwQ-32B | 79.1 | 40.9 | 63.2 | 59.6 | 38.5 | 28.8 | 8.3 | 30.1 | 61.6 | 19.3s |
| RAgent-QwQ-32B | 81.4 | 43.0 | 68.4 | 62.1 | 51.2 | 26.9 | 8.3 | 33.9 | 64.0 | 24.4s |
| *Autonomous Search within Reasoning* | | | | | | | | | | |
| Search-o1-QwQ | 84.9 | 49.5 | 73.6 | 67.2 | 53.8 | 34.6 | 16.7 | 39.8 | 67.2 | 27.6s |
| Search-R1-QwQ | 89.5 | 50.5 | 73.6 | 69.6 | 51.3 | 40.4 | 16.7 | 41.7 | 69.6 | 35.1s |
| R1-Searcher-QwQ | 88.3 | 51.6 | 73.6 | 69.6 | 53.8 | 44.2 | 16.7 | 44.6 | 74.4 | 44.5s |
| WebThinker-QwQ-RL | 90.7 | 50.5 | **78.9** | 70.7 | 56.4 | **50.0** | 16.7 | 48.5 | 74.4 | 51.0s |
| Meta-Researcher-QwQ-Base | 83.7 | 45.2 | 68.4 | 68.1 | 48.7 | 40.4 | 8.3 | 39.8 | 71.2 | 50.2s |
| Meta-Researcher-QwQ-RL | **93.0** | **53.7** | **78.9** | **73.2** | **61.5** | 48.1 | **25.0** | 50.5 | **79.2** | 45.9s |

Direct reasoning primarily relies on the model's inherent reasoning time, whereas enhancing reasoning with RAG workflow and autonomous search within reasoning involve additional search processes. Given that our method controls the length of thought content during the training phase, its reasoning time is, in fact, shorter than that of WebThinker-QwQ-RL and Meta-Researcher-QwQ-Base. The experimental results also demonstrate that Meta-Researcher achieves a favorable balance between efficiency and effectiveness, exhibiting high cost-effectiveness.

## C.6 AVERAGE TOOL INVOCATION BUDGET COMPARISON FOR DIFFERENT METHODS

Table 9 presents the average tool invocation budget of different methods. The experimental results show that although Meta-Researcher invokes search tools at a slightly higher frequency than methods like WebThinker, the difference is extremely small. This also illustrates that the performance improvement of Meta-Researcher is not achieved by substantially increasing the tool invocation budget; rather, it is realized through reinforcement learning techniques that enhance the model's planning, reflection, and reasoning capabilities, thereby achieving performance breakthroughs (i.e., more precise planning and more intelligent reflection). This is also the core idea of our paper: integrating the four core capabilities of planning, searching, reflecting, and reasoning into a single model, thereby opening up a novel training perspective for the field of in-depth research.

Table 9: Comparison of the average tool invocation budget. $ACC_L$ denotes the performance when adopting LLM-as-Judge. $S_{Turn}$ refers to the actual count of search tool invocations.

| Methods | GPQA | | GAIA | | Bamboogle | |
|---|---|---|---|---|---|---|
| | $ACC_L$ | $S_{Turn}$ | $ACC_L$ | $S_{Turn}$ | $ACC_L$ | $S_{Turn}$ |
| RAgent | 62.1 | 0.4 | 33.9 | 1.7 | 64.0 | 1.3 |
| Search-o1 | 67.2 | 0.7 | 39.8 | 2.9 | 67.2 | 2.0 |
| Search-R1 | 69.6 | 1.5 | 41.7 | 3.5 | 69.6 | 2.4 |
| R1-Searcher | 69.6 | 1.6 | 44.6 | 3.8 | 74.4 | 3.5 |
| WebThinker | 70.7 | 1.8 | 48.5 | 4.5 | 74.4 | 4.0 |
| Meta-Researcher-32B-Base | 68.1 | 0.9 | 39.8 | 3.6 | 71.2 | 3.4 |
| Meta-Researcher-32B-RL | **73.2** | 1.6 | **50.5** | 4.7 | **79.2** | 4.1 |

As virtual tools, the planning and reflection tools are entirely generated by Meta-Researcher without requiring any assistance from external tools. Therefore, we have not included them in the tool invocation budget. For the search tool, we employed the Google Search API interface provided by Serper[1], with a pricing of $0.5/1k (i.e.,it costs 0.5 dollars for 1,000 invocations).

## C.7 ABLATION STUDIES OF REINFORCEMENT LEARNING'S IMPACT

To demonstrate the necessity of reinforcement learning training, we employed the Meta-Researcher—which had undergone RL training—to construct partially fine-tuned data. We then filtered this data to retain only correctly formatted samples and fine-tuned QwQ-32B exclusively on these filtered samples. Table 10 presents a performance comparison between the Meta-Researcher trained via supervised fine-tuning (SFT) and trained through reinforcement learning (RL).

Table 10: Performance comparison between training solely with fine-tuning and training solely with reinforcement learning, $ACC_L$ denotes the performance when adopting LLM-as-Judge.

| Methods | GPQA | GAIA | Bamboogle |
|---|---|---|---|
| | $ACC_L$ | $ACC_L$ | $ACC_L$ |
| Meta-Researcher-32B-Base | 68.1 | 39.8 | 71.2 |
| Meta-Researcher-32B-SFT | 69.7 | 41.7 | 73.6 |
| Meta-Researcher-32B-RL | **74.4** | **50.5** | **79.2** |

Meta-Researcher-32B-SFT denotes the model trained exclusively via supervised fine-tuning (SFT), whereas Meta-Researcher-32B-RL indicates the model trained using reinforcement learning (RL). The results show that although SFT alone yields modest improvements over the baseline, the gains remain limited. In contrast, after our two-stage RL training, the model's performance improves substantially. This demonstrates the necessity of reinforcement learning.

SFT primarily ensures that the model adheres strictly to the prescribed output format, resulting in a modest performance gain of 1.97 points (attributable to the format itself). In contrast, reinforcement learning not only enforces standardized responses but also enhances Meta-Researcher's planning, reflection, and reasoning abilities through autonomous exploration. These capability improvements are responsible for Meta-Researcher's significant performance boost(a 7.9 points increase).

---

[1]https://serper.dev/?gad_source=1

## C.8 UNIVERSALITY OF META-RESEARCHER

To comprehensively demonstrate the universality of Meta-Researcher, we have supplemented comparative results trained using the Qwen3-14B model, with detailed findings presented in Table 11. As can be observed from the results, our method remains effective on the Qwen3 series of models, exhibiting good universality. It shows a significant improvement over direct reasoning and standard RAG approaches, and consistently outperforms the untrained Meta-Researcher-Base. This once again substantiates the effectiveness of reinforcement learning in enhancing the model's capabilities for planning, reflection, and reasoning.

Table 11: Comparative Performance of Different methods on the Qwen3-14B Model. $ACC_L$ denotes the performance when adopting LLM-as-Judge.

| **Methods** | *GPQA* | *GAIA* | *Bamboogle* | *HLE* |
| --- | --- | --- | --- | --- |
| | $ACC_L$ | $ACC_L$ | $ACC_L$ | $ACC_L$ |
| Direct Reasoning | 64.0 | 25.2 | 52.4 | 3.8 |
| Standard RAG | 63.1 | 33.9 | 63.2 | 8.0 |
| Meta-Researcher-Base | 66.7 | 36.9 | 66.4 | 12.2 |
| Meta-Researcher-RL | 71.2 | 42.7 | 74.4 | 16.0 |

## C.9 LEARNING CURVES OF META-RESEARCHER

Figure 3 illustrates the dynamic changes during the training process of Meta-Researcher. From the training curves, it can be observed that the format reward rapidly increases and stabilizes in the early stages, while the response accuracy improves slowly at first, experiences rapid growth in the mid-stages, and then maintains relative stability in the later stages. This phenomenon can be attributed to our adoption of a two-stage training approach. In the first stage, we solely focus on training the model's response format without considering response accuracy; thus, the format reward can swiftly improve during this period, and the slight increase in response accuracy mainly stems from the correctness of the format. Upon entering the second stage of training, we concurrently take into account the answer reward and the thinking length reward, with an emphasis on enhancing the model's planning quality, reflection quality, and reasoning capabilities, as well as enabling the model to learn to streamline its thinking. Consequently, the answer accuracy begins to rise during this stage, while the thinking length gradually decreases.

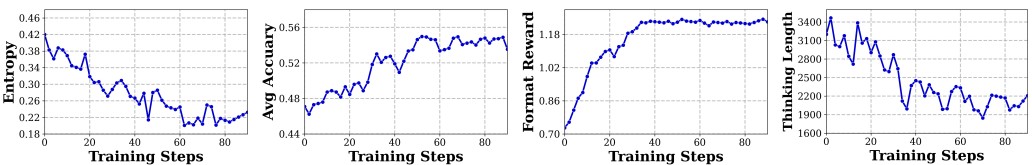

Figure 3: Learning curves of Meta-Researcher. From left to right, the curves represent the entropy curve, the average accuracy curve of benchmarks, the format reward curve, and the curve of changes in thinking length, respectively.

# D INSTRUCTION TEMPLATES

## D.1 INSTRUCTIONS FOR META-RESEARCHER

---
**Instruction for Task Planning Tool**

Task Planning Tool. It can formulate a detailed task plan for the input problem, with specific planning principles as follows:
1. For any problem, you must conduct task planning for it in your first-round reply.

---

2. A successful task plan needs to decompose the research problem into multiple sub-problems, and take into account both the breadth of the problem (covering basic aspects) and the depth (detailed information of each aspect).
3. The task plan ensures that the most critical perspectives are covered within {max_step_num} steps, with no more than {max_step_num} key points under each step.
4. Task plan needs to be output in following format:
Step 1: Collect information about xxx
    - Collect information about xxx

    ...
Step 2: Collect information about xxx
    - Collect information about xxx

    ...
...

## Instruction for Process Reflection Tool

Process Reflection Tool. It can reflect on the currently collected information, identify information gaps, and supplement information. The specific reflection requirements are as follows:
1. The process reflection stage is indispensable. After each step of the task plan is executed, it is necessary to reflect on whether there is any missing information in the current step.
2. Never be satisfied with the minimum amount of information. Unless the strictest standard of contextual sufficiency is met, it is default to collect more information.

## Instruction for Web Search Tool

Web Search Tool. This tool retrieves relevant web page information by inputting search keywords. Use the web search tool in the following situations:
1. The question exceeds your knowledge scope, and you lack the specific information needed to answer accurately.
2. The query involves the latest data, knowledge beyond your training data's cutoff time, or real-time updates.
3. More detailed or novel information is required, such as current online trends or real-time events.
4. You encounter unfamiliar terms, entities, or concepts—search instead of fabricating information.

## System Instruction for Meta-Researcher

Current time: {CURRENT_TIME}.
The language of the reply shall always be the same as that of the user's question.
# Tools
You may call one or more functions to assist with the user query. You are provided with function signatures within `<tools> </tools>` XML tags:
`<tools>`
{tool_descs}
`</tools>`
For each function call, return a json object with function name and arguments within `<tool_call> </tool_call>` XML tags:
`<tool_call>`
{"name": `<tool>`, "arguments": `<args>`}
`</tool_call>`

## User Instruction for Meta-Researcher

You are a professional AI deep researcher responsible for utilizing diverse tools to execute task planning and thoroughly solve users' problems.

# Core Task Categories
1. Formulate detailed task plans in response to users' questions.
2. Implement task plans step by step, flexibly utilizing diverse tools.
3. Collect information, verify facts, and reflect on the process.
4. Data processing, analysis, and visual presentation.
5. Compose multi-chapter articles and in-depth research reports.
6. Provide accurate answers to users' inquiries.

# Output Format Specifications
- You must first output the thinking content in `<think>...</think>` tags.
- The problem solving process must fully include the following four stages (task planning, plan execution, process reflection, problem response), with none omitted!
- Each stage must adhere to its specific format.

## **Task Planning Phase (Mandatory)**
- **Label Requirements:**: You can use the `<tool_call>...</tool_call>` tag after `<think> ...</think>` to call the **Task Planning Tool**.
- **Planning Principles**:
   - For any problem, you must first call the Task Planning Tool to formulate a task plan in your initial response.
   - A successful task plan should decompose the research problem into multiple sub-problems, addressing both the breadth (covering fundamental aspects) and depth (providing detailed information for each aspect) of the issue.
- **Format Example**:
   ```
   <think> ... </think>
   <tool_call>
   {'name': 'Plan', 'arguments': <args>}
   </tool_call>
   ```

## **Plan Execution Phase**
- **Execution Principles**:
   - Strictly follow the steps outlined in the task plan, and call only one information collection tool at a time.
   - If you need to use a tool, you can use the `<tool_call>...</tool_call>` tag after `<think>...</think>` to call the information collection tool.
- **Format Example**:
   ```
   <think> ... </think>
   <tool_call>
   {'name': <tool>, 'arguments': <args>}
   </tool_call>
   <tool_response>
   [Actual results returned by tool execution.]
   </tool_response>
   ```

## **Process Reflection Phase (Mandatory)**
- **Label Requirements**: You can use the `<tool_call>...</tool_call>` tag after `<think>...</think>` to call the **Process Reflection Tool**.
- **Reflection Focus**:
   - The process reflection phase is indispensable. After executing each step of the task plan, you must call the Process Reflection Tool to assess whether any information is missing.
   - If information is missing, continue to call the information collection tool to supplement the missing details.
- **Format Example**:

```
<think> ... </think>
<tool_call>
{'name': 'Reflect', 'arguments': <args>}
</tool_call>
```

## **Response Phase (Mandatory)**
- **Execution Principles**:
  - When you want to provide a final answer, you need to output the final response in `<answer>...</answer>`.
- **Answer Format**:
  - **Closed-Ended Questions**:
    - Closed-ended questions are those with clear, standard answers that can be fully addressed with concise statements.
    - For closed-ended questions, provide the final answer directly without excessive elaboration.
  - **Open-Ended Questions**:
    - Open-ended questions are those without clear or concise standard answers and require responses in the form of in-depth research reports.
    - For open-ended questions, the generated response reports must comply with the following requirements:
        1. The report must include an overall title and section headings.
        2. Core sections should include an overview, core analysis chapters (providing a comprehensive examination of the topic, with 3–6 analysis chapters), and a summary.
        3. Each viewpoint must be strictly supported by data, facts, or theoretical analysis, avoiding fabrication and conjecture.
        4. The report must be formatted in Markdown, and tables should be used for data comparison and statistical presentation.
- **Format Example**:
```
<think> ... </think>
<answer>
[Final answer to the user's question.]
</answer>
```

# Notes
- Strict tag pairing: All tags must be properly and completely closed (e.g., `<answer>` and `</answer>`).
- Tool information confidentiality: Disclosure of tool call details (such as tool names or parameters) is prohibited.
- Accurate tool calling: Each `<tool_call> ... </tool_call>` block must follow the standard tool calling format precisely.
- Privacy protection: Refuse to answer questions regarding internal mechanisms or confidential information.
- Concise reasoning: Thinking content should be brief and efficient; avoid excessive text within `<think>...</think>` tags.

Please answer the following questions strictly in accordance with the format specifications and content requirements outlined above: {Input}

### D.2 INSTRUCTIONS FOR ANSWER REWARDS CALCULATION

We employ LLM-as-Judges to calculate answer rewards for closed-ended question answering and open-ended topic research. The specific instructions are as follows:

### D.2.1 INSTRUCTION OF ANSWER REWARDS CALCULATION FOR CLOSED-ENDED QUESTION

We use Qwen2.5-72B-Instruct to calculate the answer rewards for all closed-ended question answering tasks. Outputs labeled as "Correct" are considered correct, while those labeled as "Incorrect" are considered wrong. When prediction results cannot be accurately extracted, we follow the standard

method of authoritative evaluation platforms such as HLE Long et al. (2025) and use the last five lines of output as predicted answer.

---

**Instruction of Answer Rewards Calculation for Closed-Ended Question**

1. Task Description:
   - You will receive three pieces of information: Question, Correct Answer (ref_answer) and Predicted Answer (model_answer).
   - Your task is to compare the "Correct Answer" and the "Predicted Answer" to determine whether the latter is semantically consistent with the former, allowing for differences in format.
   - The judgment result must be one of two types: "Correct" or "Incorrect".

2. Evaluation Requirements:
   - First, ensure you fully understand the meaning and requirements of the question.
   - Next, conduct a detailed comparison between the "Predicted Answer" and the "Correct Answer," analyzing their consistency in logic, calculations, and expression.
   - Identify and explain any errors or inconsistencies in the "Predicted Answer".
   - Finally, provide an overall judgment based on the comparison: if the answers are semantically consistent, classify the result as "Correct"; if inconsistent, classify it as "Incorrect".

3. Output Format:
   - Please carefully evaluate the predicted answer and provide the final judgment in the following JSON format:
   ```json
   {
      'Step_Evaluation': [Detail the steps for comparing answers, identify correct parts, deviations or errors, and specify the issues.],
      'Evaluation_Reasons': [Summarize the basis for your evaluation and explain the problems existing in the predicted answer.],
      'Judgment': [Correct or Incorrect.]
   }
   ```
   Below are the question, the correct answer, and the predicted answer:
   Question:{question}
   Correct answer:{ref_answer}
   Predicted answer:{model_answer}

---

### D.2.2 INSTRUCTION OF ANSWER REWARDS CALCULATION FOR OPEN-ENDED RESEARCH

For these tasks, we evaluate reports across four key dimensions: ***content completeness, information richness, consistency of data and facts, and structural rationality*** to determine the answer reward.

- *Content Completeness:* Evaluates whether the report fully addresses all key aspects of user's inquiry.

- *Information Richness:* Assesses the depth and reliability of the report's core content, ensuring it is well-supported by relevant data.

- *Consistency of Data and Facts:* Verifies the accuracy and objectivity of cited information, checking for exaggerations or unsupported claims.

- *Structural Rationality:* Examines the logical organization of the report, ensuring clarity, coherence, and alignment among sections.

We use the Qwen3-235B-A22B to calculate answer rewards through the following instructions.

**Instruction of Answer Rewards Calculation for Open-ended Research**

# Report Quality Scoring Expert: Overall Report Scoring Criteria (0-8 points)

You are a rigorous report quality scoring expert. Please evaluate the quality of the generated report based on the following four dimensions: Content Completeness, Information Richness, Consistency of Data and Facts, and Structural Rationality. Ensure that your assessment strictly adheres to the evaluation criteria below to accurately reflect the quality of the generated report.

## I. Content Completeness: Maximum 2 points, Minimum 0 points
### 1.1 **Full Coverage of Core Requirements**
- Must fully address all core requirements specified in the user's research topic; none can be omitted.
### 1.2 **Completeness of Key Modules**
- Key modules must be complete and include, but are not limited to, "Report Title," "Introduction/Background," "Analysis Content," and "Conclusion and Recommendations".

## II. Information Richness: Maximum 2 points, Minimum 0 points
### 2.1 **Multi-level Information in Core Modules**
- Core modules must contain multi-level information, including explanations, data, analysis, and conclusions.
- Content in core modules must not be excessively brief; a few sentences of analysis or general statements are unacceptable.
### **2.2 Data Support for Core Modules**
- Claims presented in core modules must be supported by factual data or evidence; content must not be superficial or overly generalized.
- All analytical content must include in-depth analysis, reasoning, and data interpretation.

## III. Consistency of Data and Facts: Maximum 2 points, Minimum 0 points
### 3.1 **Clear Data Sources**
- Data used to support claims in report must have clear and traceable sources. Data with no traceable origin may be considered false or fabricated.
### 3.2 **Authenticity of Data**
- Key information (such as data and conclusions) must align with common sense and authoritative sources; conflicting information or "obviously false" statements are unacceptable.
- Data must be factually accurate, without subjective assumptions or exaggerated expressions.

## IV. Structural Rationality: Maximum 2 points, Minimum 0 points
### 4.1 **Reasonable Directory Hierarchy**
- The report must include an overall title.
- The directory hierarchy must be logical; for example, conclusions should appear after the analysis in the core sections.
- Directory content must be relevant to the research topic and specifically address distinct aspects of it.
- Section titles must be independent and non-repetitive.
### 4.2 **Clear Article Structure**
- The article must have a clear and highly readable structure, with consistent font, font size, and line spacing for text at the same level.
- Content across different sections must be distinct, avoiding repetition or significant overlap.

## Output Format:
- Please carefully evaluate the quality of the report and provide the final score in the following JSON format:
```json
{
```

'Scores_of_Each_Dimension': [List the score for each dimension in standard Markdown format, and include detailed reasons for any deductions.],
'Overall_Score': [Final overall score for the report, ranging from 0 to 8 points, enclosed in square brackets, e.g., [5.8].]
}
"""

Please strictly assess the report quality according to the criteria specified in the four dimensions above, and ensure that the final score is provided.
User's research topic:
{topic}
Input report:
{report}

## D.3 INSTRUCTIONS FOR EVALUATION

The evaluation method for closed-ended questions can be found in the evaluation instruction shown below. For open-ended questions, to ensure fairness in evaluation, we adopt the evaluation instruction consistent with those in Xiaoxi et al. (2025b) to replace the open-ended report scoring system we used for reinforcement learning training. The specific content of the instruction can be referred to in the work Xiaoxi et al. (2025b).

---

**Instruction of Answer Rewards Calculation for Closed-Ended Question**

You are an evaluation assistant. Please determine if the predicted answer is equivalent to the labeled answer. The predicted answer does not need to be completely identical to the labeled answer, but it must be consistent in meaning and key information.
You should first provide your reasoning for the judgment, and then give your judgment result (i.e., correct or incorrect).

Question:{question}
Labeled answer:{labeled_answer}
Predicted answer:{pred_answer}

The output should in the following json format:
```json
{
    'Step_Evaluation': [Detail the steps for comparing answers, identify correct parts, deviations or errors, and specify the issues.],
    'Evaluation_Reasons': [Summarize the basis for your evaluation and explain the problems existing in the predicted answer.],
    'Judgment': [Correct or Incorrect.]
}
```

---

## E EXAMPLES OF INVOKING VIRTUAL PLANNING AND REFLECTION TOOL

Table 12 showcases specific instances of invoking the virtual planning tool and the virtual reflection tool. From these examples, it is clearly evident that within the tool invocation parameters generated by Meta-researcher, the content in the "arguments" field represents the outcomes of planning and reflection. In other words, both the planning content and the reflection content are autonomously generated by the core intelligent agent of Meta-researcher. The content within the `</tool_response>` tag fully matches that of the parameters, primarily to ensure the coherence of tool invocation. Specifically, each tool invocation necessitates a corresponding `</tool_response>` section; hence, we directly replicate the planning and reflection results generated by Meta-researcher and place them within the `</tool_response>` section as the tool's returned content. It is important to emphasize, however, that this replicated content does not participate in the loss calculation. Consequently, throughout the entire virtual tool invocation process, no

Table 12: Examples of invoking virtual planning tool and virtual reflection tool.

| Example from Bamboogle Dataset |
|---|
| **Question:** Who is the mother of the father of Barack Obama? |

| Invoking virtual planning tool |
|---|

<think> Okay, let's tackle this question: "Who is the mother of the father of Barack Obama?" Hmm, first I need to break it down step by step. [...]. Let's start with the Plan tool call. </think>
</tool_call>
{"name": "Plan", "arguments": {"plan": "Step 1: Identify Barack Obama's father's full name. - Collect the name of Barack Obama's father. Step 2: Research the mother of Barack Obama's father (Stanley Dunham) - Determine the name and background of the mother of Barack Obama's father"}}
</tool_call>
<tool_response> <plan>
Step 1: Identify Barack Obama's father's full name.
    - Collect the name of Barack Obama's father.
Step 2: Research the mother of Barack Obama's father (Stanley Dunham)
    - Determine the name and background of the mother of Barack Obama's father
</plan> </tool_response>

| Invoking virtual reflection tool |
|---|

<think> From the web search results provided, it's clear that his father is Barack Obama Sr., [...] I should call the Reflect tool to note this discrepancy and then proceed with the correct search. </think>
</tool_call>
{"name": "Reflect", "arguments": {"reflect": "The initial plan incorrectly identified Barack Obama's father as Stanley Dunham, but search results confirm his father is Barack Obama Sr. (Kenyan). The task plan must be revised to focus on Barack Obama Sr.'s maternal line. Next step: research Barack Obama Sr.'s mother."}}
</tool_call>
<tool_response> <reflect>
The initial plan incorrectly identified Barack Obama's father as Stanley Dunham, but search results confirm his father is Barack Obama Sr. (Kenyan). The task plan must be revised to focus on Barack Obama Sr.'s maternal line. Next step: research Barack Obama Sr.'s mother.
</reflect> </tool_response>

external information is introduced, and the planning and reflection content are fully integrated with Meta-researcher, allowing for optimization through reinforcement learning training.

# F CASE STUDY

The examples in Tables 13 and 14 demonstrate the effectiveness of **Empowering Planning and Reflection Mechanisms in Large Reasoning Models**, further illustrating the advancement of the Meta-Researcher method. Task planning reduces the difficulty of information collection by decomposing complex problems; meanwhile, process reflection can dynamically correct errors in the reasoning process and fill in information gaps, thereby enabling accurate and comprehensive information collection.

In the case of the Bamboogle dataset (Table 13), Meta-Researcher initially broke down the original question into two simpler sub-questions (Collect the name of Barack Obama's father; Determine the name and background of the mother of Barack Obama's father). During the reflection phase, it accurately identified errors in the current thought process (i.e., recognizing that Barack Obama's father is not Stanley Dunham) and information gaps (the search results did not contain any relevant information about Obama's grandmother). By gathering additional information, Meta-Researcher answers the final question correctly: the mother of Barack Obama's father is Habiba Akumu Nyanjango.

Additionally, we present a failed case, as demonstrated in Table 15, where the model still failed to provide an accurate answer to the question, even after triggering the reflection process and filling in the identified information gaps. In this instance, the user's query was as follows: "All of the individuals who formally held the position of United States secretary of homeland security prior to April 2019, excluding those who held the position in an acting capacity, have a bachelor's degree.

Of the universities that these bachelor's degrees were from, which is the westernmost university and which is the easternmost university? Give them to me as a comma-separated list, I only want the name of the cities where the universities are located, with the westernmost city listed first."

Although Meta-Researcher identified the information gaps pertaining to Tom Ridge's bachelor's university and Kirstjen Nielsen's bachelor's university during its second reflection and successfully supplemented the relevant information, the model failed to recognize that additional eligible individuals remained undiscovered due to the incomplete results from the initial search. Consequently, when the reflection mechanism was triggered, the model did not identify this critical information deficiency, leading to the failure to search for John F. Kelly, who also met the criteria as a secretary of homeland security, and ultimately resulting in an incorrect response.

Table 13: A closed-ended question answering example from main reasoning process of Meta-Researcher on Bamboogle dataset, where the task planning and reflection phase are highlighted using different colored fonts. [...] indicates omitted content.

| Example from Bamboogle Dataset |
|---|

**Question:** Who is the mother of the father of Barack Obama?
**Labeled Answer:** Habiba Akumu Nyanjango

| Model Output by Meta-Researcher *(True)* |
|---|

`<think>` Okay, let's tackle this question: "Who is the mother of the father of Barack Obama?" Hmm, first I need to break it down step by step. [...]. Let's start with the Plan tool call. `</think>`
`</tool_call>` {"name": "Plan", "arguments": {"plan": "Step 1: Identify Barack Obama's father's full name. - Collect the name of Barack Obama's father. Step 2: Research the mother of Barack Obama's father (Stanley Dunham) - Determine the name and background of the mother of Barack Obama's father}}
`</tool_call>`
`<tool_response> <plan>`
Step 1: Identify Barack Obama's father's full name.
  - Collect the name of Barack Obama's father.
...
To ensure the coherence of tool invocation, the parameters (i.e., the planning content) generated when Meta-Researcher calls the planning tool are directly replicated here.
`</plan> </tool_response>`
`<think>` Okay, so I need to figure out who the mother of Barack Obama's father is. [...]. `</think>`
`</tool_call>` {"name": "web_search", "arguments": {"queries": ["Barack Obama's father's full name"]}} `</tool_call>`
`<tool_response>` [...] `</tool_response>`
`<think>` From the web search results provided, it's clear that his father is Barack Obama Sr., [...] I should call the Reflect tool to note this discrepancy and then proceed with the correct search. `</think>`
`</tool_call>` {"name": "Reflect", "arguments": {"reflect": "The initial plan incorrectly identified Barack Obama's father as Stanley Dunham, but search results confirm his father is Barack Obama Sr. (Kenyan). The task plan must be revised to focus on Barack Obama Sr.'s maternal line. Next step: research Barack Obama Sr.'s mother."}} `</tool_call>`
`<tool_response> <reflect>`
Similarly, to ensure the consistency of tool invocation, the parameters (i.e., the reflective content) generated when Meta-Researcher calls the reflection tool are directly replicated here.
`</reflect> </tool_response>`
`<think>` [...] Search results corrected that to Barack Obama Sr. Next I need to focus on Barack Obama's paternal grandmother, [...] I should proceed with the next tool call as planned `</think>`
`</tool_call>` {"name": "web_search", "arguments": {"queries": ["Barack Obama's paternal grandmother"]}} `</tool_call>`
`<tool_response>` [...] `</tool_response>`
`<think>` No information about Barack Obama's grandmother was found in the provided[...] `</think>`
`</tool_call>` {"name": "Reflect", "arguments": {"reflect": "Search results did not contain any relevant information about Obama's grandmother, leaving an information gap. I should adjust the search query and continue the search."}} `</tool_call>`
`<tool_response> <reflect>` [...] `</reflect> </tool_response>`
`<think>` Search results corrected that to Barack Obama Sr. So now I need to focus on Barack Obama Sr.'s mother, [...] I should proceed with the next tool call as planned `</think>`
`</tool_call>` {"name": "web_search", "arguments": {"queries": ["Barack Obama Sr.'s mother", "Barack Obama's paternal grandmother"]}} `</tool_call>`
`<tool_response>` [...] `</tool_response>`
`<think>` From the web search results provided, it's clear that his father is Barack Obama Sr., [...] Therefore, the answer should be Habiba Akumu Nyanjango. I need to make sure [...] `</think>`
`</tool_call>` {"name": "Reflect", "arguments": {"reflect": "Web search results confirm Barack Obama Sr.'s mother is Habiba Akumu Nyanjango (Habiba Akumu Obama). All required information has been collected. No further gaps identified."}} `</tool_call>`
`<tool_response> <reflect>` [...] `</reflect> </tool_response>`
`<think>` Okay, let me wrap this up. The user asked for the mother of Barack Obama's father. [...]So the final answer should be Habiba Akumu Nyanjango. Just need to present that clearly. `</think>`
`<answer>` The mother of Barack Obama's father (Barack Obama Sr.) is **Habiba Akumu Nyanjango**. She was the paternal grandmother of Barack Obama and the second wife of Hussein Onyango Obama.
`</answer>`

1512
1513
1514
1515
1516
1517
1518
1519
1520
1521
1522
1523
1524
1525
1526

Table 14: A closed-ended question answering example from main reasoning process of Meta-Researcher w/o Plan&Reflect on Bamboogle dataset, [...] indicates omitted content.

1527

| Example from Bamboogle Dataset |
| --- |

1528
1529

**Question:** Who is the mother of the father of Barack Obama?
**Labeled Answer:** Habiba Akumu Nyanjango

1530

| Model Output by Meta-Researcher w/o Plan&Reflect *(False)* |
| --- |

1531
1532
1533
1534
1535
1536
1537

```
<think>
```
Okay, let's see. The question is asking for the mother of the father of Barack Obama. So first, I need to [...]. I should make sure to use the tool properly. The function requires an array of queries. Since the question is straightforward, maybe just one query is enough. Let me check the tool's parameters again. The instructions say to use a single search query unless multiple are needed. Since this is a direct question, one query should suffice. Alright, I'll call the web_search function with the query "Barack Obama's paternal grandmother name".
```
</think>
</tool_call>
```
{"name": "web_search", "arguments": {"queries": ["Barack Obama's paternal grandmother name"]}}
```
</tool_call>
<tool_response>
```
{"results": ["Doc0: Title: Sarah Onyango Obama - Wikipedia, Link:..., Content: Sarah Onyango Obama (1922 – March 29, 2021) was a Kenyan educator and philanthropist. She was the third wife of Hussein Onyango Obama [...]", "Doc1: Title: The Remarkable Life of Granny - Barack Obama, Link:..., Content:My family and I are mourning the loss [...]", [...]]}
```
</tool_response>
<think>
```
Okay, let me process this information. The user asked for the mother of Barack Obama's father. From the search results, the first result mentions Sarah Onyango Obama as the third wife of Hussein Onyango Obama, [...] The BBC article might be using "step-grandmother" in a different context, perhaps due to her being a later wife, but the key point is that she is widely recognized as his paternal grandmother. The answer should be Sarah Onyango Obama.
```
</think>
<answer>
```
The mother of Barack Obama's father was **Sarah Onyango Obama**. She was a Kenyan educator and philanthropist, often affectionately called "Mama Sarah" or "Granny."
```
</answer>
```

1538
1539
1540
1541
1542
1543
1544
1545
1546
1547
1548
1549
1550
1551
1552
1553
1554
1555
1556
1557
1558
1559
1560
1561
1562
1563
1564
1565

Table 15: An example of a closed-ended question-answering scenario on the GAIA dataset where Meta-Researcher triggers reflection but still provides an incorrect response, with the task planning and reflection phases highlighted using different colored fonts. [...] indicates omitted content.

| Example from GAIA Dataset |
| --- |

**Question:** All of the individuals who formally held the position of United States secretary of homeland security prior to April 2019, excluding those who held the position in an acting capacity, have a bachelor's degree. Of the universities that these bachelor's degrees were from, which is the westernmost university and which is the easternmost university? Give them to me as a comma-separated list, I only want the name of the cities where the universities are located, with the westernmost city listed first.

**Labeled Answer:** Santa Clara, Boston

| Model Output by Meta-Researcher *(False)* |
| --- |

`<think>` Okay, let me try to tackle this question. [...]. Let's start with the Plan tool call. `</think>`

`</tool_call>` {"name": "Plan", "arguments": {"plan": "Step 1: Identify all Secretaries of Homeland Security prior to April 2019 (non-acting) - List names and terms of service. Step 2: Gather bachelor's degree universities and their locations - For each Secretary, confirm undergraduate institution and city. Step 3: Determine geographical extremes - Calculate longitudinal coordinates to identify westernmost/easternmost cities"}} `</tool_call>`

`<tool_response>` `<plan>` [...] `</plan>` `</tool_response>`

`<think>` Okay, let me start by executing the first step of the plan [...]. `</think>`

`</tool_call>` {"name": "web_search", "arguments": {"queries": ["List of United States Secretaries of Homeland Security prior to April 2019 excluding acting"]}} `</tool_call>`

`<tool_response>` [...] `</tool_response>`

`<think>` Okay, Let me process the search results to identify the non-acting Secretaries of Homeland Security before April 2019, [...] Let me call the Reflect tool to identify gaps. `</think>`

`</tool_call>` {"name": "Reflect", "arguments": {"reflect": "A list of eligible non-acting ministers has been compiled, including: Tom Ridge; Michael; Chertoff; Janet Napolitano; Jeh Johnson; Kirstjen Nielsen. With no information gaps present, we can proceed to the next planning step."}} `</tool_call>`

`<tool_response>` `<reflect>` [...] `</reflect>` `</tool_response>`

`<think>` Now, the next step is to find each of their bachelor's degrees and universities. [...]. `</think>`

`</tool_call>` {"name": "web_search", "arguments": {"queries": ["Tom Ridge bachelor's degree university", "Michael Chertoff undergraduate school", "Janet Napolitano bachelor's degree university", "Jeh Johnson undergraduate degree", "Kirstjen Nielsen bachelor's degree"]}} `</tool_call>`

`<tool_response>` [...] `</tool_response>`

`<think>` [...] Wait, according to the user's instructions, after each step, I must reflect and check for missing info. Let me first compile the current data and then call the Reflect tool to identify gaps. `</think>`

`</tool_call>` {"name": "Reflect", "arguments": {"reflect": "Current information collected: - Michael Chertoff: Harvard University (Cambridge, MA) - Janet Napolitano: Santa Clara University (Santa Clara, CA) - Jeh Johnson: Partial data suggests Stanford University (Stanford, CA). Missing information: 1. Tom Ridge's bachelor's university 2. Kirstjen Nielsen's bachelor's university. Need to supplement these gaps."}} `</tool_call>`

`<tool_response>` `<reflect>` [...] `</reflect>` `</tool_response>`

`<think>` Okay, let me tackle the missing information. [...] I'll make two separate web_search calls for each of them. Let's structure the queries to get precise results. `</think>`

`</tool_call>` {"name": "web_search", "arguments": {"queries": ["Tom Ridge undergraduate degree", "Kirstjen Nielsen bachelor's university"]}} `</tool_call>`

`<tool_response>` [...] `</tool_response>`

`<think>` Alright, let's process the new search results. First, for Tom Ridge's undergraduate degree: The Britannica entry states he went to Harvard University for his B.S. in 1967. So his undergrad is Harvard (Cambridge, MA). That fills in the gap. [...] `</think>`

`</tool_call>` {"name": "Reflect", "arguments": {"reflect": "The graduation institutions of all individuals have been collected. 1. Ridge: Harvard (Cambridge, MA); 2. Chertoff: Harvard (Cambridge, MA); 3. Napolitano: Santa Clara (Santa Clara, CA); 4. Johnson: Stanford (Stanford, CA); 5. Nielsen: Georgetown (Washington, D.C.). Next, it is necessary to identify the cities where these institutions are located and determine the westernmost and easternmost cities based on longitude. No additional information supplementation is required."}} `</tool_call>`

`<tool_response>` `<reflect>` [...] `</reflect>` `</tool_response>`

`<think>` [...] Stanford is the furthest west, Cambridge is the furthest east. Therefore, the answer is: Stanford, Cambridge. [...]So yes, Stanford (city) and Cambridge (city). `</think>`

`<answer>` **Stanford, Cambridge** `</answer>`

