# OpenReview forum: "Meta-Researcher: Empowering Planning and Reflection Mechanisms in Large Reasoning Models for Advanced Deep Research Abilities"
_ICLR.cc/2026/Conference — Submitted to ICLR 2026_

### Official Review · Reviewer_N2r2 · 2025-10-25

**Soundness:** 2
**Presentation:** 3
**Contribution:** 2
**Rating:** 4
**Confidence:** 4

**Summary:**

This paper proposes Meta-Researcher, an end-to-end reinforcement learning framework that empowers Large Reasoning Models (LRMs) with explicit metacognitive capabilities for deep research tasks. The key innovation is making the planning and reflection processes controllable by explicitly outputting them rather than keeping them implicit within reasoning. The framework consists of four components: task planning, tool calling, process reflection, and question answering, optimized using Group Relative Policy Optimization (GRPO) with carefully designed rewards. Experiments on closed-ended QA (GPQA, GAIA, Bamboogle, HLE) and open-ended research (Glaive) demonstrate improvements over existing methods.

**Strengths:**

1.	Clear framework design: The four-component architecture is intuitive and the use of "virtual tools" (Task Planning Tool and Process Reflection Tool) to ensure explicit outputs is a good design choice.
2.	Comprehensive experimental evaluation: The paper evaluates on multiple diverse benchmarks covering both closed-ended and open-ended scenarios, with detailed ablation studies demonstrating the value of each component.
3.	Practical applicability: Experiments on different model scales (7B, 14B, 32B) and non-reasoning models show the framework's broad applicability.

**Weaknesses:**

1.	Limited algorithmic novelty: The paper essentially combines existing techniques (task decomposition, web search, reflection, GRPO) without introducing fundamentally new methods. The main contribution is engineering-focused rather than conceptually novel. It's primarily about formatting outputs explicitly and reward engineering.
2.	Weak theoretical foundation: (1) The proposed "metacognition" definition (Section 3) lacks rigorous justification and theoretical grounding; (2) No formal analysis of why explicit output of planning/reflection is superior to implicit reasoning; (3) The connection between the framework design and actual metacognitive capabilities is assumed rather than demonstrated.
3.	Missing critical information: (1) No computational cost analysis for training time, inference latency, API costs for web searches not reported; (2) How does the multi-round search and reflection compare to simpler approaches in terms of cost-effectiveness? (3) Limited scalability analysis: What happens with longer contexts or more complex problems?
4.	All experiments use the same web search API and similar domains (academic/factual questions).
5.	It’s unclear how the method performs with different information sources, noisy/contradictory information, or specialized domains.
6.	Some design may have problems. For example, the thinking length penalty (Equation 10) may hurt performance on genuinely complex problems.

**Questions:**

1.	Hyperparameter sensitivity: How sensitive is the method to the specific reward weights and thresholds? Was there extensive hyperparameter search, and how would practitioners set these for new tasks?
2.	Ablation on RL: What happens if you use the same explicit format with supervised fine-tuning only, without RL? Is RL necessary or is it primarily the format that helps?
3.	Information loss: How much information is lost during context truncation, and how does this affect final performance? Can you quantify this?
4.	Reflection mechanism: Why is it necessary to explicitly call a "reflection tool" rather than just prompting the model to reflect? What does the tool-calling format add?
5.	Failure modes: Can you provide systematic analysis of when the method fails? What types of questions or scenarios are still challenging?

---

> ### Author Response · Authors · 2025-11-20
> **The paper has been updated and is now accessible.**
>
> Thank you very much for your suggestions. We have provided detailed explanations for each of your comments and incorporated the supplementary experimental results into the appendix section of the paper (the newly added content is highlighted in cyan). The updated paper has been submitted and is now available for review. We will re-typeset the newly added experiments in the final version of the paper and place them in the extended pages.
>
> The comments regarding the "Weaknesses" section partially overlap with those in the "Questions" section, and we have addressed them collectively in our responses to the questions.

---

> ### Author Response · Authors · 2025-11-20
> **Response to Question 1**
>
> Response to Question 1:
>
> Thank you very much for your valuable comments. Firstly, Meta-Researcher exhibits low sensitivity to hyperparameters, and its performance does not fluctuate drastically due to minor changes in hyperparameter settings. Specifically, the parameter $\delta$, set at 1e-4, primarily serves to constrain the thinking length reward within a reasonable range. This prevents the thinking length reward from overly dominating the overall strategy update direction while encouraging the model to learn to streamline its thinking. It is recommended to adjust this parameter within the range of 1e-4 to 3e-4.
>
> Secondly, the parameter $\eta$ is mainly used to control the length of thinking content during each reasoning process for LRMs such as QwQ-32B. Through experimental testing, we set $\eta$ to 1024, 2048, 3072, and 4096, respectively. We found that excessively short thinking length lead to insufficient model reasoning, severely impacting performance, while excessively long thinking length cause the reasoning context to exceed length limitations, preventing the complete output of final answers. After balancing thinking length and performance, we ultimately selected 2048 as the thinking length threshold. However, if sufficient training resources are available, it is advisable to set the context length as large as possible during training and gradually increase the value of $\eta$ to achieve stronger performance.
>
> Lastly, the parameter $\xi$ primarily controls when the model enters the second stage of two-stage training. When $\xi$ = 1, it indicates that the model can perfectly adhere to the instruction format. At this point, the model no longer needs to rely on format reward for guidance and can proceed to the second training stage, focusing on enhancing its planning, reflection, and reasoning capabilities.

---

> ### Author Response · Authors · 2025-11-20
> **Response to Question 2**
>
> Response to Question 2:
>
> Thank you for your question! To verify the necessity of reinforcement learning (RL), we employed Meta-Researcher (trained via reinforcement learning) to generate a portion of fine-tuning data. We then filtered this data to retain only samples with the correct format, and these filtered samples were subsequently used to fine-tune the QwQ-32B model. Detailed performance comparisons between Supervised Fine-Tuning (SFT) and reinforcement learning can be found in Appendix C.7.
>
> From the comparison results, it can be observed that while model performance improved to some extent after using SFT alone compared to the baseline, the improvement was extremely limited. In contrast, after undergoing our designed two-stage reinforcement learning training, the model's performance achieved a significant leap. This fully demonstrates that reinforcement learning is indispensable. SFT can only ensure that the model strictly adheres to the instruction format for output, resulting in a performance improvement of 1.97 points, primarily attributable to format compliance. In contrast, reinforcement learning not only ensures that Meta-Researcher responds in a standardized format but also guides the model to explore autonomously, enhancing its planning, reflection, and reasoning capabilities. It is these enhanced capabilities that enabled Meta-Researcher to achieve a substantial performance improvement of 7.9 points.

---

> ### Author Response · Authors · 2025-11-20
> **Response to Question 3**
>
> Response to Question 3:
>
> For the GPQA, GAIA, and Bamboogle evaluation datasets, we respectively calculated the context truncation ratios during the response process as 1.0%, 6.8%, and 3.2%. The performance losses caused by context truncation on these three datasets were 1.0%, 0%, and 1.6%, respectively. Additionally, the issue of context truncation and its associated performance losses are more severe in open-ended evaluation datasets, with an average truncation rate of 16.9%. Therefore, we recommend expanding the model's context length as much as possible during training and inference when resources permit, to ensure substantial improvements in model performance.
>
> Although there is a truncation phenomenon in the GAIA dataset, it does not cause performance degradation. The main reason is that the truncated samples are the most challenging problems in the GAIA dataset; therefore, even if the entire response is complete, the model cannot effectively gather all relevant information and provide accurate answers.

---

> ### Author Response · Authors · 2025-11-20
> **Response to Question 3**
>
> Response to Question 3:
>
> For the GPQA, GAIA, and Bamboogle evaluation datasets, we respectively calculated the context truncation ratios during the response process as 1.0%, 6.8%, and 3.2%. The performance losses caused by context truncation on these three datasets were 1.0%, 0%, and 1.6%, respectively. Additionally, the issue of context truncation and its associated performance losses are more severe in open-ended evaluation datasets, with an average truncation rate of 16.9%. Therefore, we recommend expanding the model's context length as much as possible during training and inference when resources permit, to ensure substantial improvements in model performance.
>
> Although there is a truncation phenomenon in the GAIA dataset, it does not cause performance degradation. The main reason is that the truncated samples are the most challenging problems in the GAIA dataset; therefore, even if the entire response is complete, the model cannot effectively gather all relevant information and provide accurate answers.

---

> ### Author Response · Authors · 2025-11-20
> **Response to Question 4**
>
> Response to Question 4:
>
> I will address why it is necessary to explicitly invoke reflection tools rather than directly using prompt from the following aspects:
>
> (1) Improving the success rate of reflection output: Prior to training, we conducted experiments on both methods. When using prompt to instruct the model to generate planning and reflection content, only 11% of responses accurately generated planning and reflection processes. This resulted in consistently zero format reward during the initial training phase, making stable training difficult. In contrast, when using virtual tools, over 85% of responses correctly output planning and reflection processes. Consequently, during the early stages of reinforcement learning training, there were significant differences in advantages among different responses, guiding stable policy updates.
>
> (2) Virtual tools are more suitable for reward calculation: If reflection content is generated through prompt, we need to decode the model's output and extract the planning and reflection components to calculate rewards. This approach is overly cumbersome and may lead to unreliable rewards due to extraction failures. However, when using virtual tools, we can directly count the number of Meta-Researcher invokes planning and reflection virtual tools to calculate rewards, making reward assignment both simple and efficient.
>
> (3) Making uncontrollable planning and reflection behaviors controllable: When planning and reflection are implicit within the thinking content, these behaviors are essentially uncontrollable. We cannot guarantee whether, when, or how the model engages in planning and reflection. By using virtual tools, we can employ stable rewards during training to control the model's reflection behavior, transforming the uncontrollable planning and reflection processes into controllable ones.
>
> (4) Virtual tools do not hinder the improvement of the model's planning and reflection capabilities: Unlike introducing real external tools, the parameters for invoking planning and reflection tools are generated by Meta-Researcher itself, and these parameters represent the results of planning and reflection. Therefore, the task planning and reflection processes do not receive any information beyond the explicitly passed parameters. For planning and reflection tools, the content in <tool_response> is directly copied from the tool invocation parameters "arguments" to ensure the coherence of tool invocations, and this part does not participate in loss calculation. Thus, during the reinforcement learning process, Meta-Researcher continuously optimizes the quality of its output planning and reflection content based on feedback from response rewards, achieving self-evolution.

---

> ### Author Response · Authors · 2025-11-20
> **Response to Question 5**
>
> Response to Question 5:
>
> Thank you for your valuable comments! On the evaluation datasets of GPQA, GAIA, and Bamboogle, we hardly found any instances where reflection caused responses to change from correct to incorrect. This aligns with the prevailing research viewpoint in the current industry: the reflection mechanism has general applicability in enhancing the cognitive abilities of LLM. It not only strengthens the LLM's own fault tolerance and error correction capabilities but also demonstrates superior performance in planning for complex problems.
>
> However, we have identified instances where reflection was triggered but no valuable information was collected; specific examples are presented in Appendix F. Notably, such instances occur more frequently in the GAIA dataset and less so in GPQA and Bamboogle. A possible reason is that the GAIA dataset is more complex and demands higher precision in information collection.
>
> We have also calculated the reflection frequency metrics across the three datasets and found that the average number of planning steps is 2.4, while the average number of reflection steps is 2.7. Therefore, there is no evidence of excessive reflection. This is also related to our reward calculation method, which encourages the model to reflect thoroughly but limits the maximum number of reflections to prevent the model from stacking rewards through excessive reflection.

---

> ### Author Response · Authors · 2025-11-21
> **Response to Weaknesses 1**
>
> Response to Weaknesses 1:
>
> Thank you for your valuable feedback. Our approach offers the following advantages:
>
> (1) Currently, relevant work in Deep Research typically divides the entire research process into four modules: the planning process, the searching process, the reflection process, and the response process. These are implemented by invoking different sub-agents to carry out the complete research workflow. In contrast, the core of our method lies in integrating these four essential capabilities—planning, searching, reflection, and reasoning—into a single unified model. Through carefully designed rewards and reinforcement learning training strategies, we enhance the model’s proficiency across all four tasks, offering a novel training paradigm for deep research. Unlike prior approaches that rely on cascading multiple sub-agents, our framework achieves the desired performance using only a single backbone model, which can be continuously improved via reinforcement learning to strengthen its deep research capabilities—significantly reducing deployment complexity and cost.
>
> (2) Our method explicitly outputs the planning and reflection processes, rather than relying on prompts to implicitly embed them within the model’s internal reasoning. This explicit formulation transforms what was previously an unobservable and uncontrollable behavior into a controllable and verifiable one. With implicit prompting, there is no guarantee that the model genuinely performs planning or reflection, nor can we control when or how these cognitive steps occur. In contrast, Meta-Researcher explicitly generates planning and reflection content through virtual tool invocations. Reinforcement learning is then used to regulate both the frequency and timing of these two stages, ensuring that the model truly acquires and reliably executes these critical capabilities.

---

> ### Author Response · Authors · 2025-11-21
> **Response to Weaknesses 2**
>
> Response to Weaknesses 2:
>
> Thank you for your insightful comment, which has provided crucial guidance for improving our work. In response to your concern, we are actively conducting further investigation and incorporating cross-disciplinary theoretical foundations to strengthen the theoretical grounding of our definition of metacognition. We will also systematically clarify the connection between our framework design and metacognitive capabilities.
>
> Regarding why explicit planning and reflection are superior to implicit reasoning, we have provided a detailed explanation in our response to Question 4.

---

> ### Author Response · Authors · 2025-11-21
> **Response to Weaknesses 3**
>
> Response to Weaknesses 3:
>
> Appendix C.6 of the revised paper presents the average tool-call budget of different methods, along with the cost price of calling search APIs. Compared with methods such as R1-searcher and WebThinker, although the Meta-Researcher has a slight increase in tool calls, the difference is not significant. This indicates that it does not enhance performance by substantially increasing the number of searches. Instead, it employs reinforcement learning techniques to improve the model's planning, reflection, and reasoning capabilities, enabling more accurate planning, smarter reflection, and more efficient reasoning, thereby achieving progress.
>
> In addition, when encountering longer contexts or more complex questions, context truncation due to exceeding the maximum length may occur. Limited by training resources, we set the context length to 40,960 (covering the total length of both input and output) and implemented a series of context management measures: (1) segmenting the search content and retaining only the most relevant parts; (2) using a thinking length reward to control the model's thinking length, enabling it to learn to think concisely. However, for extremely complex questions, responses may still be interrupted due to the context exceeding the set length, which can have a certain impact on model performance. In our response to Question 3, we have presented in detail the proportion of truncated information and its impact on the final performance.

---

> ### Author Response · Authors · 2025-11-21
> **Response to Weaknesses 4 and 5**
>
> Response to Weaknesses 4 and 5:
>
> Yes, during our experiments, we consistently utilized the Google Search API interface provided by Serper, primarily due to cost and performance considerations. The pricing for Serper's Google Search API is $0.5/1k (i.e.,it costs 0.5 dollars for 1,000 invocations), and Google Search, as a widely recognized high-quality search source, covers a broader range of information sources. Furthermore, we referred to the experimental setups in the majority of Deep Research-related papers and selected benchmark test sets commonly used in the industry to measure Deep Research capabilities. Meanwhile, we sincerely appreciate your valuable suggestions. We will expand our search sources and test the performance of Meta-Researcher across various domains.

---

> ### Author Response · Authors · 2025-11-21
> **Response to Weaknesses 6**
>
> Response to Weaknesses 6:
>
> The thought-length penalty indeed has a certain impact on the model's performance on truly complex problems. However, constrained by training resources, we are unable to extend the context length further. Under such circumstances, we specifically conducted experimental tests by setting η to 1024, 2048, 3072, and 4096 respectively. The experiments revealed that excessively short thought lengths lead to insufficient model reasoning, severely affecting performance (with an average performance loss of 4%); whereas excessively long thought lengths cause the reasoning context to exceed the length limit, preventing the complete output of the final answer (with an average performance loss of 3.7%). After balancing thought length and performance, we ultimately selected 2048 as the threshold for thought length. Nevertheless, if sufficient training resources are available, it is advisable to set the context length as large as possible during training and gradually increase the value of η to achieve stronger performance.

---

### Official Review · Reviewer_FGSG · 2025-11-01

**Soundness:** 3
**Presentation:** 3
**Contribution:** 2
**Rating:** 4
**Confidence:** 3

**Summary:**

This paper introduces Meta-Researcher, a reinforcement learning (RL) framework designed to endow large reasoning models (LRMs) with explicit metacognitive abilities for autonomous research. Instead of relying on implicit reasoning within the chain-of-thought, the framework enforces a structured “Task Planning → Information Gathering → Process Reflection → Problem Solving” loop. Meta-Researcher implements two virtual tool: a task-planning tool and a process-reflection tool;  to externalize these cognitive phases, allowing the model’s internal reasoning process to become observable, rewardable, and trainable. Training is performed using Group Relative Policy Optimization (GRPO) with layered rewards (format, accuracy, and thinking-length).

**Strengths:**

1. Making planning and reflection processes explicit and controllable through virtual tools is a sensible design choice.
2. The paper evaluates on diverse tasks spanning closed-ended QA and open-ended research, with thorough comparisons against both RAG and autonomous search methods.
3. The progression from format rewards to combined rewards addresses the challenge of lacking intermediate supervision in a principled way.

**Weaknesses:**

1. The open-ended experiments rely on a small (30-sample) Glaive subset and LLM-as-judge scoring, limiting robustness.
2. We don’t see qualitative cases of when reflection fails  e.g., reflection triggers but still misses key evidence, or over-reflects and gets penalized.

**Questions:**

1. How were the specific reward weights (δ=1e-4, η=2048, ξ=1.0) chosen? How sensitive is performance to these choices?
2. Do plan/reflect stages measurably improve evidence attribution (e.g., source coverage, citation precision/recall, redundancy reduction)?
When does reflection hurt (over-reflection, topic drift)?
3. if the planning tool is not called, the overall format reward becomes 0. This is a hard gate. Did the authors observe training instability or low sample efficiency in early stages due to this?
4. The method encourages multiple tool calls and multiple reflections. What is the average tool budget per example compared to WebThinker, RAgent?

---

> ### Author Response · Authors · 2025-11-20
> **The paper has been updated and is now accessible.**
>
> Thank you very much for your suggestions. We have provided detailed explanations for each of your comments and incorporated the supplementary experimental results into the appendix section of the paper (the newly added content is highlighted in cyan). The updated paper has been submitted and is now available for review. We will re-typeset the newly added experiments in the final version of the paper and place them in the extended pages.
>
> The comments regarding the "Weaknesses" section partially overlap with those in the "Questions" section, and we have addressed them collectively in our responses to the questions.

---

> ### Author Response · Authors · 2025-11-20
> **Response to Question 1**
>
> Response to Question 1:
>
> Thank you very much for your valuable comments. Firstly, Meta-Researcher exhibits low sensitivity to hyperparameters, and its performance does not fluctuate drastically due to minor changes in hyperparameter settings. Specifically, the parameter $\delta$, set at 1e-4, primarily serves to constrain the thinking length reward within a reasonable range. This prevents the thinking length reward from overly dominating the overall strategy update direction while encouraging the model to learn to streamline its thinking. It is recommended to adjust this parameter within the range of 1e-4 to 3e-4.
>
> Secondly, the parameter $\eta$ is mainly used to control the length of thinking content during each reasoning process for LRMs such as QwQ-32B. Through experimental testing, we set $\eta$ to 1024, 2048, 3072, and 4096, respectively. We found that excessively short thinking length lead to insufficient model reasoning, severely impacting performance, while excessively long thinking length cause the reasoning context to exceed length limitations, preventing the complete output of final answers. After balancing thinking length and performance, we ultimately selected 2048 as the thinking length threshold. However, if sufficient training resources are available, it is advisable to set the context length as large as possible during training and gradually increase the value of $\eta$ to achieve stronger performance.
>
> Lastly, the parameter $\xi$ primarily controls when the model enters the second stage of two-stage training. When $\xi$ = 1, it indicates that the model can perfectly adhere to the instruction format. At this point, the model no longer needs to rely on format reward for guidance and can proceed to the second training stage, focusing on enhancing its planning, reflection, and reasoning capabilities.

---

> ### Author Response · Authors · 2025-11-20
> **Response to Question 2**
>
> Response to Question 2:
>
> Whether the response to a question is correct indirectly reflects the information source coverage and recall rate. We have calculated the contribution of the planning and reflection stages to correct responses. The planning process enables 8.4% of complex questions to be answered correctly, while the reflection process leads to correct answers for 7.8% of complex questions.
>
> On the evaluation datasets of GPQA, GAIA, and Bamboogle, we hardly found any instances where reflection caused responses to change from correct to incorrect. This aligns with the prevailing research viewpoint in the current industry: the reflection mechanism has general applicability in enhancing the cognitive abilities of LLM. It not only strengthens the LLM's own fault tolerance and error correction capabilities but also demonstrates superior performance in planning for complex problems.
>
> However, we have identified instances where reflection was triggered but no valuable information was collected; specific examples are presented in Appendix F. Notably, such instances occur more frequently in the GAIA dataset and less so in GPQA and Bamboogle. A possible reason is that the GAIA dataset is more complex and demands higher precision in information collection.
>
> We have also calculated the reflection frequency metrics across the three datasets and found that the average number of planning steps is 2.4, while the average number of reflection steps is 2.7. Therefore, there is no evidence of excessive reflection. This is also related to our reward calculation method, which encourages the model to reflect thoroughly but limits the maximum number of reflections to prevent the model from stacking rewards through excessive reflection.

---

> ### Author Response · Authors · 2025-11-20
> **Response to Question 3**
>
> Response to Question 3:
>
> Thank you for your comment. The learning curves during the training process are presented in Appendix C.9. Specifically, we did not observe training instability. In the early stages of training, the answer reward on the test set improved relatively slowly, while the format reward adherence increased rapidly. After the format reward reached the desired threshold and training entered the second stage, Meta-Researcher's performance on the test set began to steadily improve and eventually reached optimal levels. This phenomenon is closely related to our design of the planning and reflection processes as virtual tools.
>
> Why training with virtual tools is more stable, we conducted experiments comparing two approaches prior to training. When using prompt to instruct the model to generate planning and reflection content, only 11% of the responses can accurately generate the planning and reflection processes due to the model’s weak instruction following capability. This resulted in consistently zero format reward in the early stages of training, making stable training difficult. In contrast, when using virtual tools, over 85% of the responses correctly output the planning and reflection processes. Consequently, during the early stages of reinforcement learning training, there were significant differences in advantages among different responses, which guided stable policy updates.

---

> ### Author Response · Authors · 2025-11-20
> **Response to Question 4**
>
> Response to Question 4:
>
> Appendix C.6 of the updated paper presents the average tool call budget for different methods. Compared to approaches like R1-searcher and WebThinker, Meta-Researcher shows a slight increase in tool call frequency, but the difference is not significant. This indicates that its performance improvement is not achieved by substantially increasing the number of searches. Instead, it leverages reinforcement learning to enhance the model's planning, reflection, and reasoning capabilities, enabling more accurate planning, smarter reflection, and more efficient reasoning, thereby driving progress.

---

> > ### Comment · Reviewer_FGSG · 2025-11-23
> >
> > Thank you for the detailed data. I understand that perfectly quantifying every cost is challenging, and the 'average tool invocation' (S-Turn) provided in Table 9 serves as a valuable baseline estimate. However, based on the mechanism described, it is worth noting that this metric might not fully reflect the total operational cost. Specifically: Hidden Inference Costs: Since the 'Planning' and 'Reflection' phases are virtual tools generated internally, they consume significant GPU inference tokens which are not captured by the external tool count. Actual API Costs: The model can batch multiple queries into a single tool turn (e.g., 5 queries in one turn). While this counts as only 1 'Turn,' the financial cost for the Search API would actually be higher. While I accept the current estimate as a good proxy for efficiency, keeping these 'hidden' costs in mind helps us better evaluate the model's true price-performance ratio.

---

> > > ### Author Response · Authors · 2025-11-24
> > > **The actual number of search API calls and associated costs for different methods, along with the additional hidden costs of planning and reflection tools**
> > >
> > > We sincerely appreciate your response and the new comments. Regarding the actual operational costs, taking the Bamboogle dataset as an example, we have tallied the total number of actual calls to the search API during the inference process for different methods.
> > >
> > > - RAgent: 227 calls to the Google Search API, with a cost of $0.1135.
> > > - Search-o1: 375 calls to the Google Search API, with a cost of $0.1875.
> > > - Search R1: 441 calls to the Google Search API, with a cost of $0.2205.
> > > - R1-Searcher: 612 calls to the Google Search API, with a cost of $0.306.
> > > - WebThinker: 719 calls to the Google Search API, with a cost of $0.359.
> > > - Meta-Researcher-32B-Base: 593 calls to the Google Search API, with a cost of $0.297.
> > > - Meta-Researcher-32B-RL: 721 calls to the Google Search API, with a cost of $0.3605.
> > >
> > > In addition, the planning and reflection phases are generated and uniquely owned by the Meta-Researcher itself. Therefore, we have only counted the total number of tokens consumed by the planning and reflection tools on Meta-Researcher-32B-Base and Meta-Researcher-32B-RL.
> > >
> > > Meta-Researcher-32B-Base
> > > - Planning tool: Input tokens consumed: 2.4w tokens; Output tokens consumed: 1.8w tokens.
> > > - Reflection tool: Input tokens consumed: 1.6w tokens; Output tokens consumed: 5.1w tokens.
> > >
> > > Meta-Researcher-32B-RL
> > > - Planning tool: Input tokens consumed: 2.4w tokens; Output tokens consumed: 2.1w tokens.
> > > - Reflection tool: Input tokens consumed: 1.6w tokens; Output tokens consumed: 9.3w tokens.
> > >
> > > According to the official pricing of the Qwen model, the input price for calling the QwQ-32B model is $0.003/w tokens, and the output price is $0.01/w tokens. The additional consumption costs incurred by the two methods when calling the planning and reflection tools can be calculated as follows:
> > >
> > > - Meta-Researcher-32B-Base: $0.081
> > > - Meta-Researcher-32B-RL: $0.126
> > >
> > > If the additional costs are evenly amortized across each question in the Bamboogle dataset, the extra cost per question is as follows:
> > >
> > > - Meta-Researcher-32B-Base: $0.000648
> > > - Meta-Researcher-32B-RL: $0.001008
> > >
> > > We hope that the calculation of these "hidden" costs can assist you in evaluating the true cost-effectiveness of the Meta-Researcher.

---

### Official Review · Reviewer_1j81 · 2025-11-01

**Soundness:** 2
**Presentation:** 3
**Contribution:** 2
**Rating:** 2
**Confidence:** 4

**Summary:**

Deep research is a very useful tool. However, it is often difficult to control, and reasoning errors lead to gaps in the collected information. Therefore, they propose Meta-Researcher, an end-to-end RL-based Deep Research method that supports task planning, information gathering, process reflection, and, finally, problem-solving. Their structured approach makes deep research more controllable and allows RL to enhance decision-making abilities. Their experiments show that their method outperforms existing deep research recipes.

**Strengths:**

1. The authors address one of the currently most interesting applications of LLMs, deep research. While most deep research recipes are proprietary, they make their recipe publicly available.
2. The authors aim to decompose the complex deep research pipeline into  4 key components (task planning, tools calling, process reflection, question answering) that operate in two modes (closed-ended question answering and open-ended topic research, which are sensible.
3. Designed a comprehensive list of reward components, which generally seem well motivated to enable effective RL fine-tuning.
4. The benchmarks across closed-ended question answering and open-ended research are well selected.
5. The authors provided the prompts used in every part of the pipeline in the Appendix, which can be useful for future work.

**Weaknesses:**

1. Some important definitions (e.g., BGE model) or citations (GRPO) are missing
2. Generally, there is a lack of ablation studies. In particular, no sensitivity analysis of the individual reward components. Therefore, it may be that the reward function is overly complex. Can you verify that all components are necessary? Also, the two-stage training is not ablated. Is it necessary?
3. It would be valuable to see an analysis of how many steps/turns the deep research performs. Especially, how does the RL fine-tuning change this behavior? For example, does RL considerably increase the number of tool calls or result in generally longer interactions than not using RL?
4. The outcome rewards for closed-ended question answering are clear. However, there is no clarity on how the outcome of open-ended research is graded. Is LLM as a judge used on the listed dimensions? How does performance vary if you switch the judge? How reliable are the judged scores even? An analysis of this would help the paper
5. The paper focuses on Qwen 2.5-72B-Instruct. It is unclear if the findings transfer to other models. It would be essential to verify their findings with more recent reasoning models, such as Qwen3 (even if at a smaller scale), to ensure that all components and reward mechanisms are necessary. Otherwise, it is possible that the found recipe is specific to Qwen 2.5-72B.
6. It would be helpful to the reader to provide more insights into the end-to-end RL component, such as learning curves and other important metrics, which are currently missing. This also includes the computational efforts, technical challenges, etc, that come with end-to-end RL training in a difficult multi-turn problem. At the moment, it seems like RL is more used as a black-box mechanism.
7. Additionally, there is no comparison of wall-clock times of the compared methods in Table 1 (e.g., between direct reasoning, enhanced reasoning, and autonomous search). This would be important to see for users, to understand the tradeoffs of different approaches better.

**Questions:**

Some questions are asked above. Here are some more:

1. The “BGE model” is listed, but not cited or explained. Can you clarify why, what the abbreviation stands for, and how it works?
2. What is $R_{temp}^{tool}$? This is not specified.
4. Can you provide ablations on the importance of the reward components and two-stage training?
5. Can you provide a quantitative analysis of how RL changes the agent's behavior (e.g., tool call frequency, etc.)?
5. In Appendix B.1.2, the authors state that OpenResearchBench is constructed by themselves. Is this dataset going to be released?
6. How does the proposed system compare to proprietary deep research recipes from frontier labs? It is not necessary to beat these recipes, but for the reader, it would be important to see how these methods compare. If it is not feasible to conduct a full evaluation on all downstream tasks, an analysis on a restricted task set would also help.

---

> ### Author Response · Authors · 2025-11-20
> **The paper has been updated and is now accessible.**
>
> Thank you very much for your suggestions. We have provided detailed explanations for each of your comments and incorporated the supplementary experimental results into the appendix section of the paper (the newly added content is highlighted in cyan). The updated paper has been submitted and is now available for review. We will re-typeset the newly added experiments in the final version of the paper and place them in the extended pages.
>
> The comments regarding the "Weaknesses" section partially overlap with those in the "Questions" section, and we have addressed them collectively in our responses to the questions.

---

> ### Author Response · Authors · 2025-11-20
> **Response to Weaknesses 4**
>
> Response to Weaknesses 4:
>
> Thank you for your valuable comments! You are absolutely right. For open-ended research results, following the approach in the Webthinker paper, we employed the GPT-4o as a "judge" (LLM-as-a-judge) to score the reports in the test set across four dimensions. Prior to the formal evaluation, we calculated the scoring variances of different models. The results showed that both the GPT-4o and Qwen3-235b-A22B models exhibited highly stable performance, with average variances of 0.17 and 0.22, respectively. If you wish to switch the evaluation model, we recommend using either a closed-source or an open-source large-scale LLM for assessment. We once attempted to use Qwen3-32B as a scoring model to evaluate the generated reports, but found that its variance fluctuated extremely sharply, making it difficult to ensure the stability of the training process. The scoring methods and scoring prompts used for open-ended tasks are introduced in Appendices D.2 and D.3 of the paper.

---

> ### Author Response · Authors · 2025-11-20
> **Response to Weaknesses 5**
>
> Response to Weaknesses 5:
>
> In Figure 2 of the paper, we present the performance comparisons of our proposed method across different models, specifically covering DeepSeek-R1-Distill-Qwen-7B, DeepSeek-R1-Distill-Qwen-14B, and Qwen2.5-32B-Instruct. From the experimental results, it is clearly evident that our method successfully achieves effective performance improvements on the three models other than the QwQ-32B model, fully demonstrating a certain degree of generality of our approach. Additionally, we greatly appreciate your question regarding whether the effectiveness of our method can be validated on the Qwen3 series models. We have supplemented the comparison results of training using the Qwen3-14B model, and the detailed results are presented in Section C.8 of the updated paper's appendix. From these results, it can be observed that our method remains effective on Qwen3 series models, exhibiting high generality.

---

> ### Author Response · Authors · 2025-11-20
> **Response to Weaknesses 6**
>
> Response to Weaknesses 6:
>
> Thank you for your valuable suggestions! Regarding the entropy curve and reward curve during the reinforcement learning training process, we have provided detailed illustrations in Appendix C.9. Additionally, the key technical challenge of end-to-end reinforcement learning training in complex multi-round tasks lies in the context length management method. Due to limited training resources, we set the context length to 40,960 (covering the total length of both input and output) and implemented a series of context management measures: (1) segmenting the search content and retaining only the most relevant parts; (2) using a thinking length reward to control the model's thinking length, enabling the model to learn to streamline its thinking. However, when faced with extremely complex problems, responses may still be interrupted due to the context exceeding the set length, which can have a certain impact on model performance.

---

> ### Author Response · Authors · 2025-11-20
> **Response to Weaknesses 7**
>
> Response to Weaknesses 7:
>
> We have detailed the comparison of wall-clock time among direct reasoning, augmented reasoning, and autonomous search in Appendix C.5 of the updated paper. Since direct reasoning and augmented reasoning do not require reinforcement learning training, we only compare their wall-clock times during the actual reasoning phase. The wall-clock time of direct reasoning primarily stems from the model's inherent reasoning process; in contrast, RAG-augmented reasoning and autonomous search within reasoning have more complex wall-clock time structures due to the introduction of a search component. Notably, our method effectively controls the length of thinking content during the training phase, resulting in shorter actual reasoning times compared to WebThinker-QwQ-RL and Meta-Researcher-QwQ-Base. From the experimental results, it is evident that our method demonstrates high cost-effectiveness while balancing efficiency and effectiveness.

---

> ### Author Response · Authors · 2025-11-20
> **Response to Question 1**
>
> Response to Question 1:
>
> The BGE model mentioned in the text refers to bge-reranker-v2-m3 from the BAAI General Embedding series (released by the Beijing Academy of Artificial Intelligence, BAAI). Specifically, during the search phase, we employ this re-ranking model to conduct fine-grained relevance scoring on document snippets returned by search sources. Only the highest-scoring snippets are retained as contextual inputs for subsequent reasoning, preventing rapid exceeding of the maximum context length limit due to redundancy in external information.
>
> The primary reason for selecting bge-reranker-v2-m3 is its exceptional performance in multilingual, long-text, and fine-grained semantic matching tasks, making it particularly well-suited for handling diverse search results in open-domain research scenarios. Trained via contrastive learning, this model effectively captures deep semantic associations between queries and document snippets, thereby enhancing the relevance and information density of the input evidence.
>
> We acknowledge the oversight regarding the missing citation of GRPO and appreciate your correction. We have already added the citation information in the resubmitted version of the paper.

---

> ### Author Response · Authors · 2025-11-20
> **Response to Question 2**
>
> Response to Question 2:
>
> $R_{temp}^{tool}$ represents the overall format reward for tool calls during the information-gathering phase (i.e., the search phase) for each sampled path. Specifically, a sampled path may involve multiple search tool calls. For each call to a search tool, if its format aligns with expectations, we accumulate a format reward for a successful call (mentioned in the paper as 0.5 points) onto $R_{temp}^{tool}$. Ultimately, we take the average of $R_{temp}^{tool}$ based on the total number of tool calls $n$ for the current sampled path, using this average as the format reward $R_{tool}$ for the information-gathering phase of that sampled path. We will provide a detailed explanation of this notation in subsequent versions of the paper.

---

> ### Author Response · Authors · 2025-11-20
> **Response to Question 3**
>
> Response to Question 3:
>
> We have conducted supplementary experiments and presented the relevant results in detail in Sections C.2 and C.3 of the updated paper's appendix.
>
> For the ablation experiments on the two-stage training, detailed results can be found in Appendix C.2 of the re-uploaded paper. The experimental results indicate that when only training with the format reward, the performance converges at the 14th step, with only a very limited improvement compared to the untrained version. This suggests that merely making the model adhere to the instruction format does not effectively enhance its actual reasoning ability. When trained solely with the second-stage reward, the model can achieve performance comparable to that of the two-stage training, but its convergence speed is slower than that of the two-stage training, indicating that initially targeting the training of format-following ability aids the model in achieving rapid convergence.
>
> Regarding the ablation experiments on different reward components, detailed results can be found in Appendix C.3 of the re-uploaded paper. In closed-ended benchmarks, the answer reward contributes most significantly to improving the performance of Meta-Researcher. The format reward ranks second, as it increases the model's reflection frequency, thereby enabling more comprehensive information supplementation and collection. The impact of the thinking length reward is relatively minor, especially on the GPQA dataset. This is because the GPQA dataset is relatively simple, with fewer search iterations, making it less likely to encounter excessively long context truncation even without restricting the thinking length. In open-ended evaluation sets, however, the thinking length reward and the format reward become more crucial. Open-ended scenarios place greater emphasis on the completeness of report generation and the comprehensiveness of information collection. Removing the thinking length reward may result in incomplete reports due to excessively long contexts, severely affecting performance. Additionally, poor formatting can lead to failures in the planning, reflection, and search processes, compromising the comprehensiveness of information collection.

---

> ### Author Response · Authors · 2025-11-20
> **Response to Question 4:**
>
> Response to Question 4:
>
> Regarding agent interaction rounds, we conducted detailed statistics on the actual number of search tool invocations and the corresponding performance metrics of Meta-Researcher on the three test sets (GPQA, GAIA, and Bamboogle) before and after reinforcement learning training. The relevant experimental results can be found in Section C.4 of the updated paper's appendix. The results clearly demonstrate that reinforcement learning did not lead to a significant increase in the model's search frequency. The notable improvement in model performance primarily stems from the effective enhancement of the model's planning, reflection, and reasoning capabilities during the reinforcement learning training process.
>
> Meanwhile, the Tongyi team, in their research on WebSailor, pointed out that constructing cold-start data for fine-tuning the model before conducting reinforcement learning can effectively enhance the model's multi-round search capability and achieve superior results. In their experiments, the cold-start model maintained a consistently high and stable level of tool invocation throughout the reinforcement learning training period; in contrast, the model that underwent reinforcement learning directly, despite a continuous increase in invocation frequency, still exhibited a significantly lower overall level. This indicates that relying solely on the autonomous exploration of reinforcement learning makes it difficult for the model to master complex operational patterns such as ultra-multi-round searches, a conclusion that aligns with our experimental findings.

---

> ### Author Response · Authors · 2025-11-20
> **Response to Question 5**
>
> Response to Question 5:
>
> Thank you for the reviewer's attention. Yes, the OpenResearchBench dataset will be publicly released alongside the paper. We plan to make the dataset open-source after the paper is accepted to facilitate further exploration and reproduction by the community in the direction of deep research agents.

---

> ### Author Response · Authors · 2025-11-20
> **Response to Question 6**
>
> Response to Question 6:
>
> We compared Meta-Researcher against WebDancer, WebShaper, and WebSailor—three models specifically developed for deep research by Alibaba Tongyi Lab—on the GAIA test set. Unlike Meta-Researcher, these three methods constructed high-quality cold-start data and fine-tuned the model before conducting reinforcement learning training. Since Tongyi Lab has not open-sourced the specific training code, we are unable to replicate their performance solely using reinforcement learning training. Below, we present a performance comparison of these four methods under two scenarios: with cold-start (SFT+RL) and without cold-start (RL):
>
> - WebDancer:          SFT+RL  (51.5)         RL     (-)
> - WebShaper:          SFT+RL  (52.4)         RL     (-)
> - WebSailor:            SFT+RL   (55.4)         RL    (42.0)
> - Meta-Researcher: SFT+RL   (-)              RL     (50.5)
>
> The numbers in parentheses represent the performance of each method. From the comparison results, it is evident that Meta-Researcher has certain disadvantages compared to the other three methods. However, WebSailor has released results solely from reinforcement learning training, and under fair comparison conditions, Meta-Researcher has achieved a certain lead over WebSailor. Most crucially, our method focuses on enhancing the model's intrinsic capabilities by integrating the four core abilities—planning, searching, reflecting, and reasoning—into a single model, thereby opening up a new training perspective for the field of in-depth research, rather than focusing solely on constructing high-quality datasets.

---

### Official Review · Reviewer_uoUv · 2025-11-01

**Soundness:** 2
**Presentation:** 1
**Contribution:** 2
**Rating:** 2
**Confidence:** 4

**Summary:**

The paper describes a system, a prompting strategy and tool-calling harness to build a deep research system. The authors emphasize that the system uses planning, tool-calling, using search to gather information and reflection as crucial components. Secondly, the authors use end-to-end RL with GRPO to optimize the overall system.

**Strengths:**

Building a useful system to solve challenging tasks such as “deep research” on top of LLM remains a challenge and deserves careful attention from the research community. Tuning these systems end-to-end with RL is a promising direction and public research can have high impact.

**Weaknesses:**

My biggest concern is around the training/evaluation methodology for the results involving RL: It seems there is no distinction between training and test tasks and I am concerned that RL optimization might have been performed directly on the test problems?

Furthermore, there are very few ablations describing the impact of individual design choices of the overall system. For example: What is the impact of the two training stages described in 4.4.4? I.e. which gains come from training the model to adhere to the correct formatting vs actually improving the models capabilities?



Minor comment:

The paper frequently employs excessive adjectives and overly complex terminology. While individual instances might be justifiable, the overall impression is one of pretentiousness. For example, terms like "virtual tools," "autonomous reflection," "autonomous search," and "multi-tool collaborative calling mechanism" effectively describe one LLM initiating sub-LLM calls with predefined prompts.

I also don’t understand why running RL with reward only for adhering to the toll-calling format is “genuinely instill(ing) metacognitive capabilities for autonomous reflection” (line 310).

**Questions:**

Do you use distinct training/test tasks for RL? Do you train one model across all benchmark sets and how to you mix the differently sized datasets?

What exactly is the difference between “closed ended” and “open ended” mode? I infer that open-ended tasks use a LLM-as-judge to score the result. Figure 1 suggests open-ended is using additional tools and more reflection iterations? Lines 190 to 205 are not clearly expressing the difference. “integrating external information with the reasoning capabilities” seems to be necessary for both.

When the meta-researcher triggers a call to the task-planning or process-reflection tools, do the sub-agents receive more than the explicitly passed arguments? (the JSON tool-call syntax “{name: .. arguments: }” suggest arguments and intermediate results are passed explicitly only). The example on page 22 however suggests that the reflection tool is “inline” and has full access to the meta-researchers token stream?

How are the planning and reflection tools sub-trajectories handled for the RL updates? Are they doing their own GRPO update steps? What exactly comprises a group then?

B.2 Implementation details mentions $s_p = 6$ for the reward calculation in the “Process Reflection Phase” What is this reward and how is it used? Just returned as a number to the meta-agent?

---

> ### Author Response · Authors · 2025-11-20
> **Response to Weaknesses 2**
>
> Thank you very much for your suggestions. We have provided detailed explanations for each of your comments and incorporated the supplementary experimental results into the appendix section of the paper (the newly added content is highlighted in cyan). The updated paper has been submitted and is now available for review. We will re-typeset the newly added experiments in the final version of the paper and place them in the extended pages.
>
> The comments regarding the "Weaknesses" section partially overlap with those in the "Questions" section, and we have addressed them collectively in our responses to the questions.
>
> Response to Weaknesses 2:
> We conducted additional experiments to investigate the specific benefits contributed by each of the two training stages, and present the results in Section C.2 of the appendix in the revised version of the paper. When only training with the format reward, the performance converges at the 14th step, with only a very limited improvement compared to the untrained version. This indicates that merely making the model adhere to the instruction format does not effectively enhance its actual reasoning ability. When trained solely with the second-stage reward, the model can achieve performance comparable to that of the two-stage training, but its convergence speed is slower than that of the two-stage training. This suggests that initially targeting the training of format-following ability aids the model in achieving rapid convergence. The first-stage model focuses on format adherence, while the second-stage model focuses on enhancing planning, reflection, and reasoning abilities. The two stages have distinct roles yet collaborate with each other, making it easier for the model to quickly identify the optimization direction compared to mixed training with both format and response rewards.

---

> ### Author Response · Authors · 2025-11-20
> **Response to Question 1**
>
> We sincerely appreciate your inquiry. We have made a clear distinction between training tasks and testing tasks. During the training process, we employed the open-source ORION dataset, along with our self-constructed open-ended question training dataset, OpenResearchBench (which will be publicly released later), for reinforcement learning optimization. These two datasets were mixed in a 1:1 ratio, and we trained only a single model to evaluate its performance across all benchmark datasets. For the testing tasks, we utilized mainstream datasets including GPQA, GAIA, Bamboogle, HLE, and glaive to validate the model's performance. Detailed descriptions of each dataset are provided in Section B.1 of the appendix. Therefore, we did not directly optimize the model on the testing questions.

---

> ### Author Response · Authors · 2025-11-20
> **Response to Question 2**
>
> Response to Question 2:
>
> The primary differences between closed-ended and open-ended modes are as follows:
>
> (1) Closed-ended questions have standard answers. During reinforcement learning training, accurate and stable rewards can be obtained based on rules or the "LLM-as-judge" approach. In contrast, open-ended questions lack standard responses and require the use of LLMs for scoring according to predefined criteria. Due to the absence of labeled answers and the subjective nature of evaluation, the rewards exhibit a certain degree of noise and fluctuation.
>
> (2) To ensure comprehensive and accurate reporting, the open-ended mode involves more search iterations and denser information acquisition. Figure 1 implies that open-ended questions have more reflection iterations because their overall search process is generally longer, necessitating multiple reflections to supplement information. Of course, for exceptionally complex closed-ended questions, there may also be special cases where the search process is lengthy.
>
> (3) Regarding tool usage, both modes solely employ web search tools. We apologize for any misleading implications in the figure. Our intention was to indicate that users can extend the toolset to achieve better information collection. We regret any inconvenience caused by the misunderstanding, and the inappropriate implication in Figure 1 is corrected in the updated version of the paper.
>
> (4) Finally, as you correctly pointed out, integrating external information with reasoning capabilities is indeed essential for both modes. The focus of both modes lies in collecting external information and integrating it for reasoning, with the main differences lying in the quantity of information collected and the method of reward assignment.

---

> ### Author Response · Authors · 2025-11-20
> **Response to Question 3**
>
> Response to Question 3:
>
> Thank you for your valuable feedback. When Meta-Researcher invokes task planning and process reflection tools, there are no actual sub-agents involved (hence they are referred to as virtual tools in the paper). Meta-Researcher generates tool invocation parameters on its own, with the content in the "arguments" representing the planning and reflection results. Essentially, this approach is the same as using prompts to guide Meta-Researcher in generating planning and reflection content, but it adopts a more convenient and efficient manner. Therefore, the task planning and reflection processes do not receive any information beyond the explicitly passed parameters. For the planning and reflection tools, the content in <tool_response> is directly copied from the tool invocation parameters "arguments" to ensure the coherence of tool invocation, and this part is not involved in loss calculation. We have also revised and highlighted the unclear descriptions in Chapter 4 of the paper. We sincerely apologize for any inconvenience caused by the misunderstanding. Furthermore, we provide detailed examples of the tool invocation parameters for planning and reflection in Appendix E.
>
> The reason for designing them as virtual tools is as follows: We found that through prompts alone, the model could not perfectly follow instructions (with only 11% of responses fully outputting the planning and reflection processes), resulting in the inability of the policy to accurately identify the direction for format optimization during the early stages of reinforcement learning training. Therefore, we leveraged the tool invocation capability injected during the pre-training of the Qwen series models to generate planning and reflection content via tool invocation, enabling over 85% of responses to fully output both parts and facilitating targeted reinforcement in subsequent reinforcement learning.
>
> Additionally, this form is more convenient for reinforcement learning training. When using prompts to guide the model in generating content, rules or other methods must be employed during training to ensure that the model truly outputs planning and reflection content for reward calculation, which is unreliable and complex. However, by treating the planning and reflection processes as virtual tools, we only need to count the number of times Meta-Researcher invokes these two tools, which is convenient and stable.

---

> ### Author Response · Authors · 2025-11-20
> **Response to Question 4**
>
> Response to Question 4:
>
> The planning and reflection tools do not undergo separate GRPO updates independently. As described in the response to the previous question, the parameters for the planning and reflection tools are generated by Meta-Researcher itself, and these parameters represent the outcomes of planning and reflection. The returned results of the tools are directly copied from the parameters to ensure the coherence of tool invocation, without any involvement of external information. Therefore, they can self-update during the reinforcement learning process by enhancing format reward and answer reward, continuously generating higher-quality planning and reflection content.

---

> ### Author Response · Authors · 2025-11-20
> **Response to Question 5**
>
> Response to Question 5:
>
> During the process reflection stage, we employ format reward to regulate the number of generated process reflections. Our expectation is that Meta-Researcher will reflect on the search content after completing each planning step to determine whether all key points mentioned in the plan have been covered with information collection. If there are information gaps, a new round of information supplementation will be triggered. Therefore, Appendix B.2 mentions the maximum number of steps that can be planned in the planning process, and the actual number of planning steps will be used when calculating the format reward for Process Reflection Phase. This reward aims to encourage the number of reflections to be as close as possible to the number of planning steps, while also setting an upper limit to prevent Meta-Researcher from excessive reflection to accumulate reward.
>
> This reward mechanism is one of the key aspects of our approach. Reinforcement learning with this reward can effectively increase the frequency of Meta-Researcher's reflections, ensuring that the model learns to reflect in a timely manner and that the reflection behavior is controllable, rather than being implicit in the thinking process where we cannot regulate the model's reflection behavior and timing.

---

> ### Author Response · Authors · 2025-11-20
> **Conclusion**
>
> Finally, we sincerely apologize for any inconvenience caused by our presentation and description styles that have hindered your understanding. We will continuously optimize and polish the expressions in the methodology section of the paper to present our approach in a clearer and more intuitive manner, ensuring its flawless presentation in the final version.

---

### Comment · Area_Chair_xHHB · 2025-11-25
**Author-Reviewer Discussion Phase**

Hi Reviewers,

Remember to check the rebuttal from the author and it would be great if the discussion can continue. The deadline is Dec. 3rd.

Best,
AC

---

### Author Response · Authors · 2025-12-01
**Appreciation and Overall Summary.**

We sincerely thank all reviewers and AC for their time, constructive feedback, and insightful suggestions. We have carefully addressed every comment and revised the manuscript accordingly. **A improved version of the paper has now been uploaded**. We kindly invite the reviewers and AC to read our responses in conjunction with the revised manuscript, as this will provide clearer context and help illustrate how each concern has been resolved.

In summary, our approach focuses on **enhancing the intrinsic capabilities of the model by integrating four core abilities—planning, searching, reflection, and reasoning—into a single model**, thereby providing a novel training perspective for deep research domains. It is a widely recognized viewpoint in most studies that improving the model's self-reflection capability can further enhance its reasoning ability and the accuracy of its responses.

However, we believe that enabling the model to output planning and reflection processes during its thinking constitutes an implicit and uncontrollable state. This method cannot guarantee that the model will consistently output these two components in every response, nor can it effectively control the frequency of the model's reflections. Therefore, we have designed the planning and reflection processes as virtual tools. On one hand, this allows for precise execution of planning and reflection processes by leveraging the model's inherent tool-calling capability. On the other hand, we can regulate the number of executions and the timing of planning and reflection processes by controlling the invocation of these virtual tools. Moreover, since the parameters and returned content of both virtual tools are generated by the Meta-Researcher itself without introducing any external information, continuous improvement in the quality of planning and reflection by the Meta-Researcher can be achieved through reinforcement learning training, ultimately leading to better performance.

---

### Meta-Review · Area_Chair_m5ni · 2026-01-07

**Summary:**

This paper is about training models to perform deep search tasks. The authors introduce some extra modules like task planning and reflection as virtual tools for the models to call, and propose a two-stage training method to train the model to achieve good performance. The reviewers generally have four concerns: (1) missing ablation results for the new introduced components; (2) small open-ended set for LLM-judge scoring; (3) Limited method novelty; (4) some missing citations and details such as the cost. I have read the reviews, papers, and authors' responses. I think the concern is the missing ablations on various components which may be over-complex and over-designed. The added experiments in the rebuttal only touch the effects of two-stage training and RL itself, without finer-grained analysis of planning, reflection components. Moreover, the two-stage training results in Appendix C.2 suggests that the two-stage training may not be necessary and stage 2 only is equally good. The authors mention that two-stage training converges faster without actual empirical evidence. Therefore, the potentially over-complex components are not well-justified.

**Reviewer Concerns:**

The reviewer concerns on clarification are well-addressed such as the cost analysis and train/test overlap. Also, the concerns on ablations of RL and two-stage training are mitigated. However, the concerns on detailed ablation of introduced components and novelty are still outstanding.

**Reviewer Scores:**

First, none of the reviewers have replied to change their score.

Reviewer uoUv (2): One big concern of this reviewer was potential train/test leakage and missing ablations. The authors’ have clarified that and added the two-stage ablation results, although the two-stage results may not well support the method but suggest one-stage training is equally good. The reviewer's concerns on detailed ablations of each component remain.

Reviewer 1j81 (2 -> 4?): Many concrete requests were addressed (judge stability, generality including Qwen3-14B, plus added training/efficiency reporting). I expect this could shift them to a more borderline stance, but the remaining uncertainty about whether the reward design is overly complex is still there.

Reviewer FGSG (4): This reviewer was already near-threshold and focused on robustness/analysis details (open-ended subset size, qualitative failure cases, and tool/cost budget). The revision’s additional analyses and the ensuing cost discussion suggest their major issues were only partially addressed.

Reviewer N2r2 (4): The critique centered on limited novelty/theory and missing cost/scalability reporting. The authors add some cost proxy reporting and truncation discussion, but the novelty/theoretical issues are only acknowledged rather than resolved.

---

### Decision · Program_Chairs · 2026-01-26

Reject